# High-order Interactions Modeling for Interpretable Multi-Agent Q-Learning

**Qinyu Xu**
School of Management and Engineering
Nanjing University
qinyuxu@smail.nju.edu.cn

**Yuanyang Zhu**[†]
School of Information Management
Nanjing University
yuanyangzhu@nju.edu.cn

**Xuefei Wu**[†]
School of Management and Engineering
Nanjing University
xuefeiwu@smail.nju.edu.cn

**Chunlin Chen**
School of Robotics and Automation
Nanjing University
clchen@nju.edu.cn

## Abstract

The ability to model interactions among agents is crucial for effective coordination and understanding their cooperation mechanisms in multi-agent reinforcement learning (MARL). However, previous efforts to model high-order interactions have been primarily hindered by the combinatorial explosion or the opaque nature of their black-box network structures. In this paper, we propose a novel value decomposition framework, called Continued Fraction Q-Learning (QCoFr), which can flexibly capture arbitrary-order agent interactions with only linear complexity $\mathcal{O}(n)$ in the number of agents, thus avoiding the combinatorial explosion when modeling rich cooperation. Furthermore, we introduce the variational information bottleneck to extract latent information for estimating credits. This latent information helps agents filter out noisy interactions, thereby significantly enhancing both cooperation and interpretability. Extensive experiments demonstrate that QCoFr not only consistently achieves better performance but also provides interpretability that aligns with our theoretical analysis.

## 1 Introduction

Cooperative multi-agent reinforcement learning (MARL) has recently drawn much attention [1], such as autonomous vehicles [2, 3, 4], robotics [5, 6], and autonomous warehouses [7, 8]. Success in these domains generally depends on a tapestry of local interactions: agents influence one another not only through explicit pairwise effects but also through higher-order dependencies that shape emergent behaviour [9, 10], which is a key factor to make coordination efficient. How to model those interactions among agents and to do so in a way humans can understand remains a challenge.

Value decomposition MARL has emerged as a powerful framework with the large capacity of deep neural networks and the centralized training and decentralized execution (CTDE) paradigm, which factorises a team return into individual utilities. Leading value decomposition models—such as the monotonic mixer in QMIX [11]—capture interaction structure only implicitly, leaving their decisions opaque. Post-hoc explainers (e.g., Shapley value attribution [12] and masked-attention visualisation [13]) offer limited insight and could not recover the underlying temporal or relational dynamics. Another inherent interpretable technique, like decision tree [14, 15, 16] or Shapley-based [12], handles only low-order interactions; higher-order terms still explode combinatorially.

---

[†]Corresponding author: Yuanyang Zhu and Xuefei Wu.

39th Conference on Neural Information Processing Systems (NeurIPS 2025).

Here, we argue that modeling high-order collaboration patterns among agents is still crucial for promoting coordination, which is still missing. Continued fractions [17, 18] provide a natural remedy: their recursive form captures interactions of any order, and a simple truncation yields a finite and interpretable approximation without combinatorial blow-up.

Building on these ideas, we propose Continued Fraction Q-Learning (QCoFr), a novel framework that combines expressive value decomposition with intrinsic interpretability. QCoFr represents the joint action-value function as a weighted sum of continued fraction modules—recursive structures that explicitly approximate arbitrary-order interactions among agents with linear complexity $\mathcal{O}(n)$ in the number of agents. To strengthen credit assignment, QCoFr incorporates a variational information bottleneck (VIB) [19] to capture task-relevant latent information to bridge local histories to the optimal joint action $\boldsymbol{u}$. The assistive information will be used to estimate the credits together with global state $\boldsymbol{s}$ for assisted value factorization, which relieves the spurious correlation between state $\boldsymbol{s}$ and $Q_{tot}$. We also give a rigorous proof that QCoFr takes a linear combination of continued fractions and has the property of universal approximation, even each with finite depth and linear layers. We demonstrate through extensive experiments on LBF, SMAC, and SMACv2 benchmarks that QCoFr not only generally achieves better performances on solving all tasks but also provides interpretability lacking in the state-of-the-art baselines.

## 2   Background

**Dec-POMDP.** A fully cooperative multi-agent task can be modeled as a Decentralized Partially Observable Markov Decision Process (Dec-POMDP) [20]. A Dec-POMDP can be defined as a tuple $\langle N, S, U, P, r, Z, O, n, \gamma \rangle$, where $N$ denotes a set of $n$ agents and $s \in S$ is the global state of the environment. At each time step, agent $i$ selects an action $u_i \in U$, forming a joint action $\boldsymbol{u}_t \in \mathbf{U} \equiv U^n$. This results in the next state $s'$ according to the state transition function $P\left(s' \mid s, \boldsymbol{u}\right) : S \times \mathbf{U} \times S \to [0, 1]$. All agents obtain the same joint reward $r(s, \boldsymbol{u}) : S \times U^n \to \mathbb{R}$ and $\gamma \in (0, 1]$ is the discount factor. Due to partial observability, each agent $i \in N$ receives individual observation $z_i \in Z$ from observation function $o_i \in O(s, i)$. Each agent maintains an action-observation history $\tau_i \in T$, conditioned by its policy $\pi_i$. The overall objective is to find an optimal joint policy $\boldsymbol{\pi} = \langle \pi_1, \ldots, \pi_n \rangle$ to maximize the joint value function $Q^{\boldsymbol{\pi}}\left(s_t, \boldsymbol{u}_t\right) = \mathbb{E}\left[R_t \mid s_t, \boldsymbol{u}_t\right]$.

**Value Decomposition.** Value decomposition has emerged as a dominant paradigm in multi-agent reinforcement learning (MARL) under the centralized training and decentralized execution (CTDE) paradigm. It seeks to approximate the joint action-value function $Q_{tot}$ by decomposing it into individual utility functions $Q_i$, where each utility depends solely on the agent's local trajectory $\tau_i$. To ensure that decentralized action selection remains consistent with the globally optimal joint action, the decomposition must satisfy the Individual-Global-Max (IGM) principle [11]:

$$\arg\max_{\boldsymbol{u}} Q_{tot}(\boldsymbol{\tau}, \boldsymbol{u}) = \left\{ \arg\max_{u_1} Q_1\left(\tau_1, u_1\right), \cdots, \arg\max_{u_n} Q_n\left(\tau_n, u_n\right) \right\}. \qquad (1)$$

Among representative methods, VDN [21] implements a simple additive decomposition, treating all agents' contributions equally by summing their utilities. To capture more complex coordination while preserving IGM, QMIX [11] introduces a mixing network that combines utilities into the joint value through a monotonic function. During training, the mixing network and agent networks are jointly optimized by minimizing the temporal-difference (TD) loss of $Q_{tot}$, while during execution, agents act independently using their local policies derived from the learned utility functions. The introduction of representative related works for the above formulation can be referred to in Appendix A.

## 3   High-order Interactions Modeling for Decomposition

In a large multi-agent task, agents are usually decomposed into several coalitions, each consisting of multiple agents that cooperate to accomplish the common goal. Besides, each agent could belong to different coalitions at different time steps. This coalition organization is general and can characterize most coordination patterns among agents. Thus, it is essential to model complex interactions among agents and estimate their credits for understanding their coordination patterns. In this section, we provide an informal comparative analysis of using the continued fraction network (CFN) architecture against widely adopted value decomposition methods: VDN [21], QMIX [11], and NA²Q [22].

From the view of value factorization, VDN provides a simple yet highly interpretable approach by representing the joint Q-value simply as the sum of individual agent Q-values, $Q_{tot}(\boldsymbol{\tau}, \boldsymbol{u}) =$

$\sum_{i \in N} Q_i(\tau_i, u_i)$. VDN inherently models only first-order interactions, which could limit its ability to capture richer and group-level coordination. QMIX enriches the functional class of factorisation than that of VDN by introducing a state-conditioned monotonic mixer $Q_{tot}(s, \boldsymbol{\tau}, \boldsymbol{u}) = f_{\mathbf{QMIX}}(s, [Q_1(\tau_1, u_1), \ldots, Q_n(\tau_n, u_n)])$, where $f_{\mathbf{QMIX}}$ is constrained to be monotonic in each argument. This implicit higher-order modeling increases representational capacity, but at the expense of interpretability—$f_{\mathbf{QMIX}}$ behaves as a black box, obscuring which interactions drive performance. NA$^2$Q tries to expand the joint value via a Taylor-like decomposition across all agents up to order $n$:

$$Q_{tot} = f_0 + \sum_{i=1}^{n} \alpha_i \underbrace{f_i(Q_i)}_{\text{order-1}} + \cdots + \sum_{k \in \mathcal{D}_h} \alpha_k \underbrace{f_k(Q_k)}_{\text{order-}h} + \cdots + \alpha_{1\ldots n} \underbrace{f_{1\ldots n}(Q_1, \ldots, Q_n)}_{\text{order-}n}, \tag{2}$$

where $f_k \in \{f_1, \cdots, f_{1\ldots n}\}^m$ is modeled with neural additive model [23] and $\mathcal{D}_h$ is the set of all non-empty subsets of $h \in \{1, ..., n\}$ with order-$h$ interactions. While this yields interpretability, low-order interaction terms (e.g., 2-order), the number of terms grows exponentially, $\mathcal{O}(2^n)$, making full expansion impractical for large teams.

To overcome the limitations of existing value decomposition methods discussed above, we enrich with the value factorization via a continued fraction neural network (CFN) inspired by CoFrNets [18], whose recursive structure and linear composition form a function class with universal approximation capability [24]. Following the terminology of CoFrNets, we call the function depicted in *Fig. 1* a "ladder", as its pictorial representation resembles a rail-and-rung structure that sequentially propagates inputs through the nodes. The joint action-value function can be established as a weighted summation of individual values:

$$Q_{tot} = \sum_{k=1}^{l} \alpha_k \cdot \tilde{f}_k(\boldsymbol{Q}), \tag{3}$$

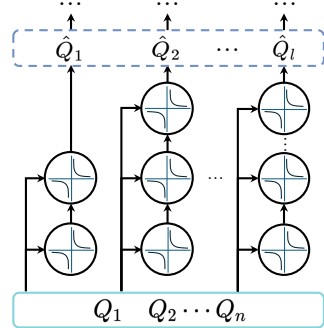

Figure 1: A diagram of CFN composed of different depth ladders that take individual values $\boldsymbol{Q}$ as inputs and output temporal values $\hat{Q}$ with $d$-order interactions.

where $\tilde{f}_k$ denotes a continued fraction structure $\frac{1}{\boldsymbol{w_{k_1}}\boldsymbol{Q} + } \frac{1}{\boldsymbol{w_{k_2}}\boldsymbol{Q} + \cdots}$ with linear functions $\boldsymbol{w_k}$ of the input $\boldsymbol{Q} = \{Q_1, \ldots, Q_n\}$ given $l$ ladders, and $\alpha_k$ are learnable weights used to estimate the contribution of individual agents and coalitions of agents. With a power series expansion, such a recursive structure can be rewritten as

$$\frac{1}{\boldsymbol{w_1}\boldsymbol{Q} + } \frac{1}{\boldsymbol{w_2}\boldsymbol{Q} + \cdots} = \sum_{p_1, \ldots, p_n = 0}^{\infty} c_{p_1, \ldots, p_n} \prod_{i=1}^{n} Q_i^{p_i}, \tag{4}$$

where the coefficients $c_{p_1, \ldots, p_n}$ and the weight parameters $\boldsymbol{w_k}$ are in one-to-one correspondence [17, 18], and each $c_{p_1, \ldots, p_n}$ used to derive interactions among agents.

CFN generally approximates interactions up to a finite depth $d$, which constitutes a rational approximation $R_d(\boldsymbol{Q})$ of the joint utility function up to $d$-order interactions among agents. It shows that truncating the continued fraction provides a practical rational approximation that explicitly models agent interactions up to order $d$:

$$\hat{Q}_k = \frac{1}{\boldsymbol{w_1}\boldsymbol{Q} + } \frac{1}{\boldsymbol{w_2}\boldsymbol{Q} + \cdots} \frac{1}{\boldsymbol{w_d}\boldsymbol{Q}} = \sum_{p_1, \ldots, p_n = 0}^{d} c_{p_1, \ldots, p_n} \prod_{i=1}^{n} Q_i^{p_i}. \tag{5}$$

**Definition 1** (Padé Approximant [25, 26]). *Let $C(z) = \sum_{k=0}^{\infty} c_k z^k$ be a formal power series in the variable $z$, then the Padé approximant of order $[L/M]$ is a rational function of the form:*

$$R_{L,M}(z) = [A_{L,M}(z)]/[B_{L,M}(z)], \tag{6}$$

*where $A_{L,M}(z)$ and $B_{L,M}(z)$ are polynomials of degrees at most $L$ and $M$, respectively, chosen such that*

$$B_{L,M}(z)C(z) - A_{L,M}(z) = \mathcal{O}(z^{L+M+1}), \tag{7}$$

*where notation $\mathcal{O}(z^k)$ denotes some power series of the form $\sum_{n=k}^{\infty} \tilde{c}_n z^n$. This approximation minimizes the difference between the rational function and the power series up to the order $L + M$.*

We give a formal proof that such truncation naturally satisfies the Padé approximation condition as defined in *Definition 1*, i.e., $f(\boldsymbol{Q}) - R_d(\boldsymbol{Q}) = \mathcal{O}(\boldsymbol{Q}^{d+1})$, where $f(\boldsymbol{Q})$ denotes the complete representation of agent interactions based on their individual value functions. This implies that the

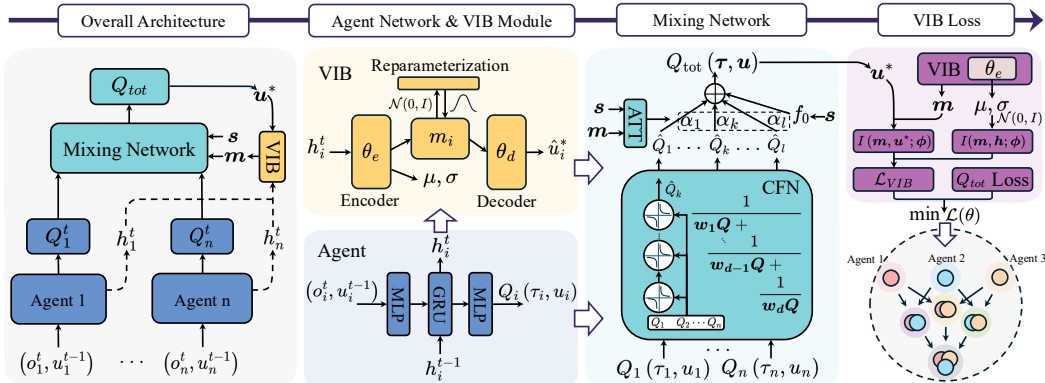

Figure 2: The overall architecture of QCoFr. A CFN-based mixing network models high-order interactions among agents by expressing $Q_{\text{tot}}$ as a linear combination of $l$ ladders over individual values $\boldsymbol{Q}$. The VIB module encodes the hidden state $\boldsymbol{h}$ into assistive information $\boldsymbol{m}$, which is used to deduce the credits of each agent of coalitions together with the global state $\boldsymbol{s}$.

depth-$d$ truncation of the continued fraction yields an exact approximation of $f(\boldsymbol{Q})$ up to the $d$-th order of $\boldsymbol{Q}$. The detailed proof is provided in Appendix B.2.

Moreover, we can easily control the capacity of the interaction order among agents with the acceptable complexity $\mathcal{O}(n)$ by adjusting the depth of CFN. Since linear combinations of CF modules constitute a function class with universal approximation power [24], they can approximate any continuous mapping between finite-dimensional spaces. Its recursive and interpretable structure reveals the contributions of individual agents and coalitions of agents, enabling precise credit assignment while faithfully modeling high-order interactions among agents. We also study two CFN variants, CFN-C and CFN-D (see *Fig. 10* and Appendix C.6). CFN-C combines single-feature and full-input CF ladders, trading higher expressivity for additional computational cost. CFN-D retains only single-feature ladders, yielding a purely additive mixer that cannot represent higher-order interactions. Empirical comparisons are provided in Section 5.2.

## 4 Methods

We next present an interpretable value decomposition framework that explicitly models arbitrary-order agent interactions without incurring a combinatorial explosion. The overall architecture of QCoFr is illustrated in *Fig. 2*, which consists of three components as follows: (1) an individual action-value function for each agent, (2) an assistive information generation module that encodes the hidden state $\boldsymbol{h}$ via a variational information bottleneck (VIB) [19] to produce the assistive information $\boldsymbol{m}$ to promote the credit assignment, and (3) a joint action-value function $Q_{tot}$ that composes individual action-value functions into a joint action-value function using a CFN architecture, where the assistive information $\boldsymbol{m}$ and the global state $\boldsymbol{s}$ are fed into the mixing network to estimate credits. During the centralized training, the whole network is learned in an end-to-end fashion to minimize the sum of TD and VIB loss, and each agent selects actions using its Q-function based on local action-observation history in a decentralized fashion.

**Individual Action-value Function** is computed by a recurrent Q-network with gated recurrent unit (GRU) for each agent $i$, which takes the local observation $o_i^t$, historical information $h_i^{t-1}$, and previous action $u_i^{t-1}$ as inputs and outputs local $Q_i(\tau_i, u_i)$.

**Assistive Information Generation Module.** Previous works usually use all available information for estimating the credit assignment $\alpha_k$, which could be quite inefficient [27]. The global state $\boldsymbol{s}$ is an unobserved confounder as the common cause factor of the global state and the joint value function [28, 22]. Thus, it will be necessary to introduce additive information to cut off the confounder. To this end, we generate an assistive latent representation $\boldsymbol{m}$ from the hidden state $\boldsymbol{h}$ of agents to predict the optimal action selection $\boldsymbol{u}^*$, which forms the Markov dependency $\boldsymbol{h} \to \boldsymbol{m} \to \boldsymbol{u}^*$. Here, we utilize VIB to capture the task-relevant information that can provide a more accurate estimation of the optimal joint action selection from the hidden state $\boldsymbol{h}$, which can promote the credit assignment. The goal is to learn an encoding that contains as little redundant information as possible while

providing maximal information about the prediction $\boldsymbol{u}^*$ that affects environmental information or its private properties, which can be written as

$$J_{IB}(\boldsymbol{\phi}) = I(\boldsymbol{m}, \boldsymbol{u}^*; \boldsymbol{\phi}) - \beta I(\boldsymbol{m}, \boldsymbol{h}; \boldsymbol{\phi}), \tag{8}$$

where the Lagrange multiplier $\beta \geq 0$ controls the tradeoff. Here, we adopt variational approximations to simplify the computation of the mutual information $I$ in Eq. 8. The following theorems provide tractable bounds for each term.

**Theorem 1** (Lower Bound for $I(\boldsymbol{m}, \boldsymbol{u}^*; \boldsymbol{\phi})$). *Let the representation $m_i$ be reparameterized as a random variable drawn from a multivariate Gaussian distribution $m_i \sim \mathcal{N}(f_m(h_i; \phi_m), \boldsymbol{I})$, where $f_m$ is an encoder parameterized by $\phi_m$, $h_i$ denotes the hidden state of agent $i$, and $\boldsymbol{I}$ is the identity covariance matrix. Then, the mutual information between the assistive information $\boldsymbol{m}$ and the optimal joint action $\boldsymbol{u}^*$ is lower-bounded as:*

$$I(\boldsymbol{m}, \boldsymbol{u}^*; \boldsymbol{\phi}) \geq \frac{1}{N} \sum_{i=1}^{N} \mathbb{E}_{\epsilon \sim p(\epsilon)} \left[ -\log q(u_i^* \mid f(h_i, \epsilon)) \right], \tag{9}$$

*where $q(u_i^* \mid m_i)$ is a variational distribution approximating the true posterior $p(u_i^* \mid m_i)$, and $m_i = f(h_i, \epsilon)$ denotes a deterministic function of $h_i$ and the Gaussian random variable $\epsilon$.*

**Proof sketch:** This bound follows from the variational representation of mutual information. By introducing a variational approximation $q(u_i^* \mid m_i)$ and applying the reparameterization trick [29] to write $p(m_i \mid h_i)dm_i = p(\epsilon)d\epsilon$, the intractable posterior is replaced with a more manageable form, enabling efficient optimization via gradient-based methods.

**Theorem 2** (Upper Bound for $I(\boldsymbol{m}, \boldsymbol{h}; \boldsymbol{\phi})$). *Let $\tilde{q}(\boldsymbol{m})$ denote a variational approximation of the marginal distribution $p(\boldsymbol{m})$. Then, the mutual information between the representation $\boldsymbol{m}$ and the hidden state $\boldsymbol{h}$ admits the following upper bound:*

$$I(\boldsymbol{m}, \boldsymbol{h}; \boldsymbol{\phi}) \leq \mathrm{KL}(p(\boldsymbol{m} \mid h_i) \,\|\, \tilde{q}(\boldsymbol{m})). \tag{10}$$

**Proof sketch:** This result exploits the non-negativity of the KL divergence. By approximating the marginal distribution $p(\boldsymbol{m})$ with a variational approximation $\tilde{q}(\boldsymbol{m})$, the mutual information can be bounded from above via the definition of the KL divergence.

Combining the above conclusions, we can arrive at the following variational objective function:

$$\mathcal{L}_{VIB} = \frac{1}{N} \sum_{i=1}^{N} \mathbb{E}_{\epsilon \sim p(\epsilon)} \left[ -\log q\left(u_i^* \mid f(h_i, \epsilon)\right) \right] + \beta \mathrm{KL}\left[ p(\boldsymbol{m} \mid h_i), \tilde{q}(\boldsymbol{m}) \right]. \tag{11}$$

The detailed derivation and rigorous proofs for these bounds are provided in Appendix B.1.

**Mixing Network.** We introduce the CFN-based mixing network consisting of $l$ ladders (*Fig. 2*) with a maximum depth $d$, which are designed to model $d$-order interactions among agents. Each ladder $k$ takes the local value functions $\boldsymbol{Q}$ as input and produces an output $\hat{Q}_k$ as defined in Eq. 5. To satisfy the IGM principle [30], each CFN layer adopts a strictly non-negative activation function of the form $\frac{1}{\max(|z|, \delta)}$, where $z$ denotes the sum of the current weighted input and the reciprocal of the previous layer's output, and $\delta$ is a small positive constant to prevent poles caused by near-zero denominators. Since the universal approximation theorem applies to any linear combination of continued fractions and does not rely on non-negative weights, an extension to non-IGM mixers is possible, with details discussed in the Appendix E.2.

Effective coordination hinges on precisely deducing the contributions of each individual agent or coalition to the overall success. To this end, we leverage assistive information $\boldsymbol{m}$ together with the global state $\boldsymbol{s}$ to estimate the credits. Specifically, the credit $\alpha_k$ for each ladder $k$ is computed as

$$\alpha_k = \frac{\exp\left( (\boldsymbol{w}_m \boldsymbol{m})^\top \mathrm{ReLU}\left( \boldsymbol{w}_s \boldsymbol{s} \right) \right)}{\sum_{k=1}^{l} \exp\left( (\boldsymbol{w}_m \boldsymbol{m})^\top \mathrm{ReLU}\left( \boldsymbol{w}_s \boldsymbol{s} \right) \right)}, \tag{12}$$

where the weights $\boldsymbol{w}_m$ and $\boldsymbol{w}_s$ are the learnable parameters, and ReLU is the activation function. When the credits $\alpha_k$ are then used to aggregate the outputs $\hat{Q}_k$ of the ladders, the joint action-value function $Q_{tot}$ can be represented as

$$Q_{tot} = \sum_{k=1}^{l} \alpha_k \hat{Q}_k = \sum_{k=1}^{l} \alpha_k \sum_{p_1,\dots p_n=0}^{d} c_{p_1\dots,\dots p_n} \prod_{i=1}^{n} Q_i^{p_i} = \sum_{p_1\dots p_n=0}^{d} c'_{p_1\dots,\dots p_n} \prod_{i=1}^{n} Q_i^{p_i}, \tag{13}$$

where $c'_{p_1,\ldots,p_n}$ denotes the final coefficient of each interaction term. For notational simplicity when discussing interaction patterns, we denote the coefficient of a specific agent interaction (e.g., agent $i$ and $j$) as $\beta_{ij}$, where the subscript indicates the agent indices involved.

**Overall Learning Objective.** To sum up, we train QCoFr end-to-end with two terms of loss functions. The first one is naturally the original mean-squared temporal-difference (TD) error, which enables each agent to learn its individual agent policy by optimizing the joint-action value of the mixing network module. The last one is the VIB loss $\mathcal{L}_{VIB}$ that is encouraged to produce assistive information. Thus, the overall loss function is formulated as follows:

$$\mathcal{L}(\theta) = \sum_i \left(Q_{tot}(s, \boldsymbol{\tau}, \boldsymbol{u}) - y_i\right)^2 + \mathcal{L}_{VIB}, \tag{14}$$

where $y_i = r + \gamma \hat{Q}_{\text{tot}}\left(s', \boldsymbol{\tau}', \arg\max_{\boldsymbol{u}' \in \mathcal{U}^n} Q_{\text{tot}}\left(s', \boldsymbol{\tau}', \boldsymbol{u}'\right)\right)$ with $\theta$ denotes the parameters of the target network. We summarize the full pseudo-code of QCoFr in Appendix C.1.

## 5 Experiments

In this section, we conduct experiments to evaluate QCoFr on three challenging benchmarks over the Level Based foraging (LBF) [31], the StarCraft Multi-Agent Challenge (SMAC) [32] and SMACv2 [33]. The details of the environment can be found in Appendix C. We compare our method against nine prominent baselines, including VDN [21], QMIX [11], QPLEX [34], Centrally-Weighted QMIX (CW-QMIX) [35], CDS [36], SHAQ [37], GoMARL [38], ReBorn [39], and NA$^2$Q [22]. We carry out experiments with 5 random seeds, and all performance results are plotted using mean ± std. Furthermore, we perform interpretability analyses of QCoFr to provide empirical evidence for the contributions of individual agents and coalitions of agents.

### 5.1 Performance Comparison

**Performance on LBF.** We first evaluate QCoFr on two constructed LBF tasks: 3 agents with 3 food items (*lbf-3-3*) and 4 agents with 2 food items (*lbf-4-2*). LBF is a grid-based multi-agent benchmark that supports configurable levels of cooperation and observability [40]. Each agent navigates a $10 \times 10$ grid world and observes a $2 \times 2$ sub-grid centered around it, where a group of adjacent agents can successfully collect a food item if

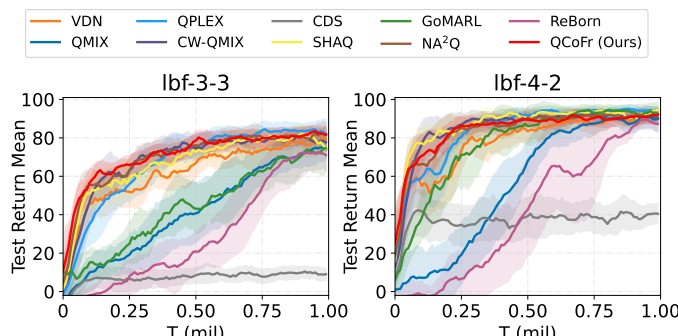

Figure 3: Performance results on LBF benchmark.

the sum of their levels is greater than or equal to the item's level. As shown in *Fig. 3*, QCoFr achieves competitive performance compared to state-of-the-art methods. However, the simplified task design inherently constrains the model's capacity to fully leverage higher-order interaction mechanisms, whose potential advantages may emerge in complex multi-agent coordination scenarios. The failure of the CDS may be due to an overemphasis on behavioral diversity without considering effective coordination among agents. QMIX has a slow convergence speed due to its inability to exploit latent information, which restricts its representational capacity. GoMARL and ReBorn also require more steps to discover effective agent groupings and allocate neuron weights. While CW-QMIX, NA$^2$Q, and SHAQ demonstrate competitive performance by introducing additional modules to enhance value decomposition, their inability to model higher-order interactions or reliance on complex computation may hinder scalability in complex multi-agent coordination tasks.

**Performance on SMAC.** We next compare QCoFr with baselines on the more challenging SMAC benchmark, where agents make decisions based on local observations while cooperating to defeat AI-controlled enemies. We show the performance comparison on six different scenarios, including one easy map: *2s3z*, three hard maps: *2c_vs_64zg*, *3s_vs_5z*, *5m_vs_6m*, and two super-hard maps: *3s5z_vs_3s6z*, *6h_vs_8z*. As shown in *Fig. 4*, QCoFr achieves superior performance across almost all

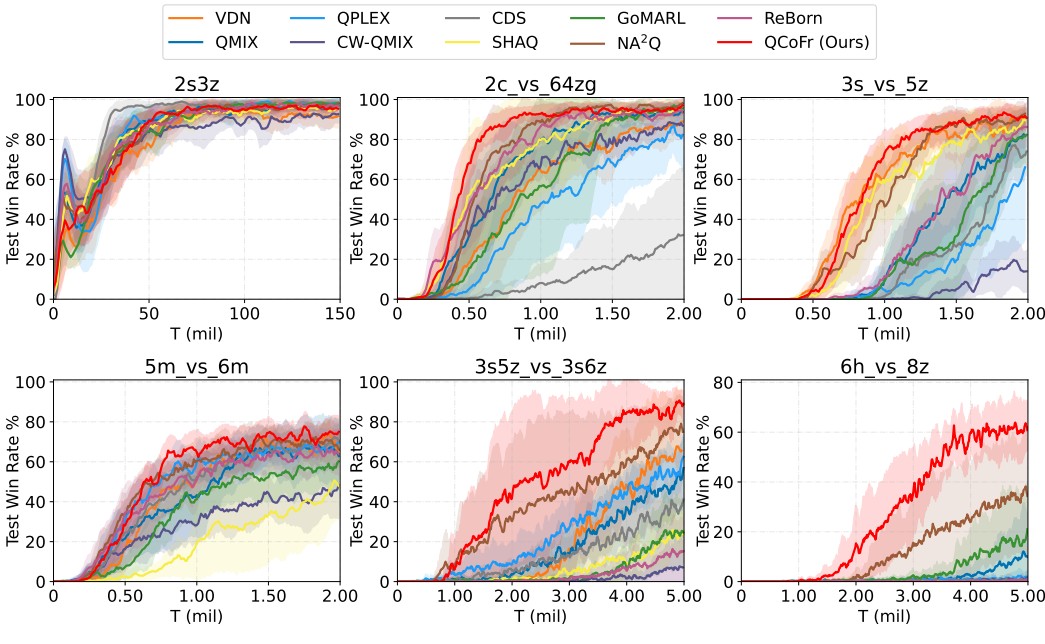

Figure 4: Performance results on SMAC benchmark.

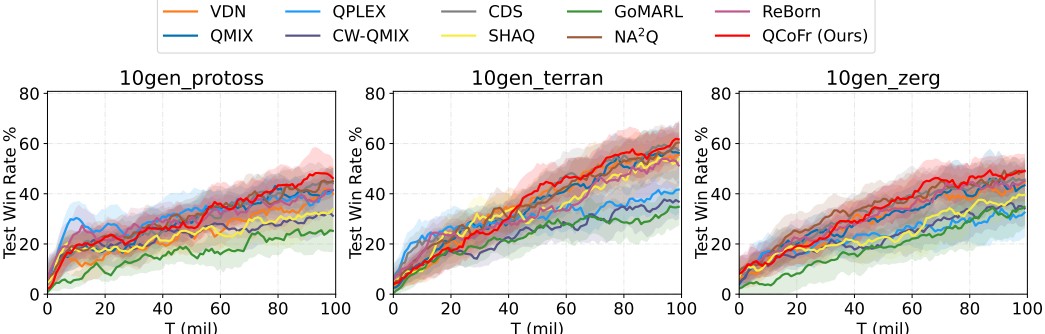

Figure 5: Performance results on SMACv2 benchmark.

scenarios, especially on the super hard tasks. CW-QMIX and QPLEX do not achieve satisfactory performance, which may be due to excessive approximation and the relaxed constraints introduced during training. CDS exhibits slower convergence rates, which may be due to requiring more training steps to capture diverse individualized behaviors. Although GoMARL and $NA^2Q$ can model high-order interactions to yield notable performance on complex scenarios by leveraging grouping and enumeration, GoMARL requires extended training durations to learn effective groupings, and $NA^2Q$ considers low-order interaction terms to avoid combinatorial explosion.

**Performance on SMACv2.** We evaluate QCoFr performance on three scenarios from SMACv2, including *zerg_5_vs_5*, *protoss_5_vs_5*, and *terran_5_vs_5*. SMACv2 introduces randomly generated and positioned for unit, increasing environmental stochasticity compared to SMAC. The performance of QCoFr is significantly better than other algorithms across all scenarios. In contrast, GoMARL performs the worst, which is due to its dynamic grouping structure, leading to slow convergence. While SHAQ demonstrates marginally superior performance, its inability to model higher-order agent interactions limits its adaptability to all maps. Compared to $NA^2Q$, QMIX, and CDS, QCoFr achieves better performance, which should benefit from the assistive information for estimating credit assignment, especially for the high-order interaction patterns among agents.

## 5.2 Ablation Studies

To discuss the influence of each component, we conduct ablation studies about (a) the number of interaction orders among agents, (b) the CFN structure, and assistive information on performance.

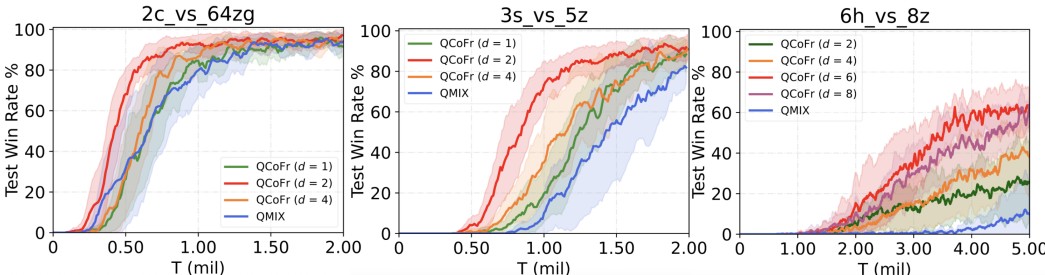

Figure 6: Comparison of different numbers of interaction orders.

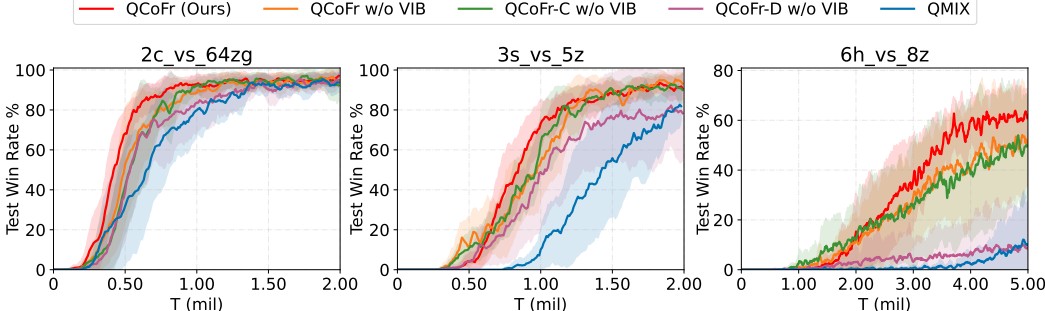

Figure 7: Performance with and without VIB and comparison of different CFN structures.

**The Number of Interaction Orders.** We conduct ablation experiments by varying the depth $d$ of CFN, which governs the highest order of inter-agent interactions modeled by the framework. On *2c_vs_64zg* and *3s_vs_5z* maps, *QCoFr ($d$ = 2)* yields significant performance improvements compared to *QCoFr ($d$ = 1)*, which underscores the importance of explicitly modeling higher-order inter-agent effects as shown in *Fig. 6*. However, beyond the optimum, performance could degrade, likely due to overfitting to spurious higher-order correlations. The same trend emerges in the *6h_vs_8z* scenario, where the model achieves its best performance with depth $d$ = 6 before degrading with further depth increases. These findings demonstrate that modeling higher-order interactions yields performance improvements for complex coordination tasks, but only within a bounded regime that aligns with the task's intrinsic interaction complexity. Notably, our method using a single-layer CFN still outperforms QMIX, which confirms that the assistive information can better help deduce the contribution of agents for their team.

**Impact of CFN Structure and Assistive Information Module.** To validate the effectiveness of the CFN structure, we compare *QCoFr w/o VIB* against two CFN-based variants—*QCoFr-C w/o VIB* and *QCoFr-D w/o VIB*, under a unified setting without the VIB module. As shown in *Fig. 7*, *QCoFr w/o VIB* achieves comparable performance to *QCoFr-C w/o VIB*, while offering a simpler structure and lower computational cost. In contrast, *QCoFr-D w/o VIB* exhibits significantly degraded performance, especially on the *6h_vs_8z* scenario, which underscores the necessity of explicit inter-agent interaction modeling for complex tasks. To summarize, we adopt CFN as the backbone for its ability to capture higher-order interactions with a simple, efficient design. By integrating a VIB-based assistive information generation module, QCoFr achieves significant improvements in both convergence speed and final performance, demonstrating its efficacy and practical utility. More additional ablations in Appendix E.1 detail the respective contributions of the VIB and CFN modules.

### 5.3 Interpretability

To intuitively demonstrate the interpretability of QCoFr, we demonstrate *QCoFr ($d$ = 2)* and *QCoFr ($d$ = 4)* and display some key frames on *5m_vs_6m* scenario. As shown in *Fig. 8(b)*, the pairwise coalitions (agent 3, agent 4) and (agent 3, agent 5) emerge as coordinated coalitions that focus fire on the same enemy unit. Here, we find that they have higher coalition contributions of $3.884$ and $3.339$ than the others. In contrast, agent 1 disengages from combat when it has low health value to avoid early elimination, which indicates that it does not contribute to the team and obtains a lower contribution of $0.552$. These observations highlight QCoFr's ability to facilitate diverse role

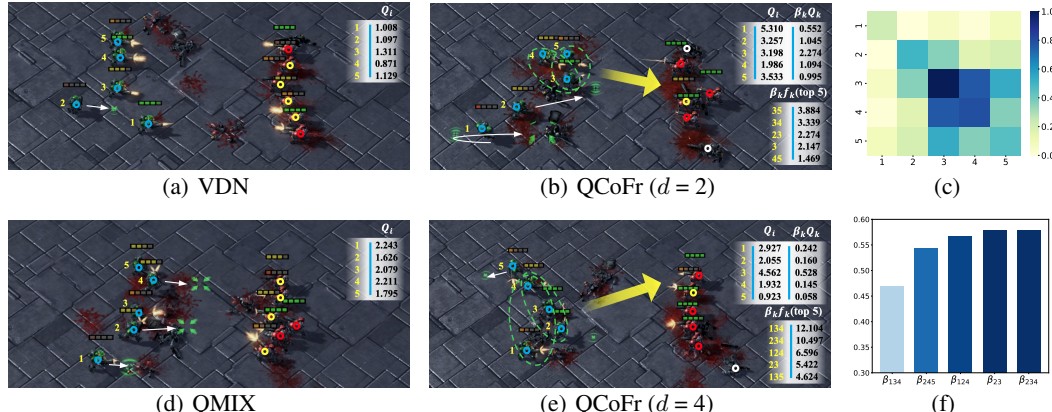

(a) VDN      (b) QCoFr ($d = 2$)      (c)

(d) QMIX      (e) QCoFr ($d = 4$)      (f)

Figure 8: Visualization of evaluation results for QCoFr and baselines on *5m_vs_6m* map. (b) and (e) illustrate the behaviors of QCoFr with CFN depths of 2 and 4 at a specific time step. (c) visualizes the weights of individual agents and pairwise coalitions corresponding to the behavior shown in (b), while (f) presents the top five highest-weighted coalitions extracted from (e), due to the increased number of possible interactions. (a) and (d) show the behaviors of VDN and QMIX for comparison.

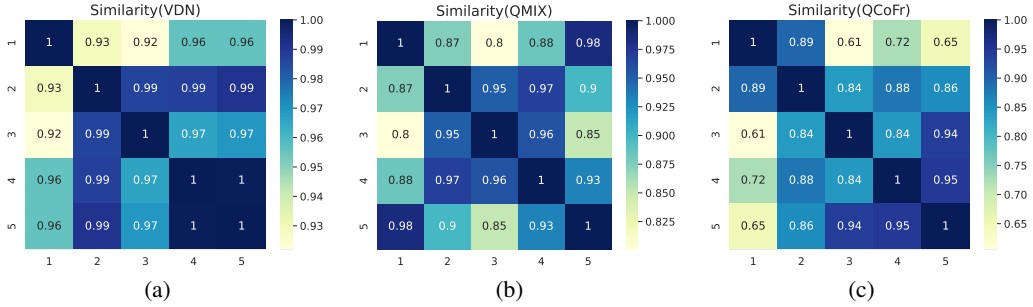

(a)      (b)      (c)

Figure 9: Visualization of agent diversity via QCoFr and baselines. Cosine similarities of agent Q-values under the same observation indicate that QCoFr yields more diverse agent behaviors.

specializations and demonstrate the advantage of modeling agent interactions, which helps deduce the contribution of individual agents and coalitions within their team.

Further, we show the behavior of the agents when $d = 4$ for QCoFr to demonstrate that the model captures complex agent cooperation as shown in *Fig. 8(e)*. As shown in *Fig. 8(c)*, the weights are predominantly concentrated on agent 3 and agent 4, as well as their interactions with other agents, consistent with their coordinated attacks. While in deeper QCoFr, top-ranked terms are predominantly higher-order, such as interactions among agents 1, 3, and 4, which yield the highest contributions, demonstrating the model's capacity to encode intricate cooperative dynamics. In contrast, QMIX and VDN produce less differentiated policies, with individual Q-values remaining close, making their decision logic harder to interpret. Furthermore, as shown in *Fig. 8(a)* and *Fig. 8(d)*, agents under VDN and QMIX frequently attack multiple enemy units simultaneously, leading to prolonged numerical disadvantage and increased failure risk.

To evaluate whether QCoFr facilitates more diverse agent behaviors during training, we compute the cosine similarity of individual Q-values under the same observation across different methods, as shown in *Fig. 9*. QCoFr yields consistently lower similarity, which indicates more diverse and thus more specialized agent preferences, in line with the qualitative findings in *Fig. 8*. In contrast, VDN and QMIX maintain similarity close to 1, which reflects homogenized preferences that hinder the emergence of complex cooperative strategies and obscure individual decision-making.

## 6   Conclusion

In this paper, we introduce QCoFr, an interpretable value-based MARL framework grounded in the expressive and compact structure of continued fractions. By leveraging continued fraction neural networks and a variational information bottleneck over agent histories, QCoFr explicitly models agent

interactions of arbitrary order while maintaining low model complexity and inherent interpretability. Extensive experiments show that QCoFr matches or surpasses strong value-decomposition baselines and yields clearer attributions to individuals and coalitions. We believe QCoFr presents a promising direction for designing MARL algorithms with mathematically grounded, interpretable structures and highlights the importance of modeling higher-order coordination. Future work will explore adaptive mechanisms to dynamically adjust the depth of interaction modeling in response to task complexity.

**Limitations.** In the current implementation, the CFN depth is fixed per task, which may be suboptimal or wasteful. A promising direction is to adapt depth during training (or per state) to balance computation with representational power.

## 7   Acknowledgements

This work was supported in part by the China Postdoctoral Science Foundation under Grant Number 2025T180877, the Jiangsu Funding Program for Excellent Postdoctoral Talent 2025ZB267, the National Key Research and Development Program of China under Grant 2023YFD2001003, Major Science and Technology Project of Jiangsu Province under Grant BG2024041, and the Fundamental Research Funds for the Central Universities under Grant 011814380048.

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

# A   Related Work

**Value Decomposition.** Centralized training with decentralized execution (CTDE) has emerged as a powerful paradigm in MARL [41, 42], where global information can be accessed during centralized training and learned policies are executed with only local information in a decentralized way. Under the CTDE paradigm, value decomposition methods show their strength in expressing the joint value function conditioned on individual value functions. VDN [21] introduces a linear decomposition, representing the joint Q-value as a sum of agent-wise Q-values. However, its additive nature ignores inter-agent interactions, limiting its scalability to complex coordination tasks. QMIX [11] improves representational capacity by employing a nonlinear monotonic mixing network parameterized via hypernetworks, but the imposed monotonicity constraint hinders its flexibility. To overcome this, QTRAN [30] introduces a relaxed transformation-based decomposition to bypass monotonicity, while WQMIX [35] incorporates a weighted projection to enhance approximation quality. QPLEX [34] further refines the decomposition by adopting a duplex dueling architecture that satisfies the Individual-Global-Max (IGM) principle via an advantage-based formulation. Despite their improvements in expressiveness, these methods primarily focus on functional accuracy and provide little insight into the underlying coordination structure. This lack of interpretability becomes particularly problematic in partially observable and interaction-intensive environments, where understanding agent dependencies is crucial for robust credit assignment. To address this, we propose a novel interpretable value decomposition framework that explicitly encodes high-order interactions, offering both performance and transparency.

**Interpretable MARL.** Recent advances in interpretable MARL can be broadly categorized into two paradigms, focusing either on (i) intrinsic interpretability or (ii) post-hoc explanation [43]. Intrinsic interpretability requires the learned model to be self-understandable by nature, which is achieved by using a transparent class of models, whereas post-hoc explanation entails learning a second model to explain an already-trained black-box model. Post-hoc methods provide auxiliary insights without modifying the underlying learning process. For instance, SQDDPG [12] estimates individual agent contributions via Shapley Q-values, while Goto et al. [13] use masked-attention to identify salient observation regions in multi-vehicle coordination tasks. Although informative, these techniques lack robustness guarantees and struggle to recover the relational or temporal structure intrinsic to multi-agent cooperation. In contrast, intrinsically interpretable approaches seek to construct models whose decision logic is understandable by design. Tree-based architectures such as MIXRT [15] and MAVIPER [14] represent agent policies using soft or symbolic decision trees, providing explicit reasoning paths. DTPO [16] advances this line by directly optimizing tree structures via policy gradients, combining transparency with performance. Attention-based models, such as MAAC [44], further enhance interpretability by dynamically identifying inter-agent dependencies, while other methods promote explainability through latent skill inference [45] or constrained policy spaces that encode global objectives [46].

Within the value decomposition framework, central to cooperative MARL, several works also try to understand how agents cooperate via agent-level contributions. VDN [21] factorizes the team reward additively, assuming agent independence. SHAQ [37] adopts Shapley value theory to quantify marginal contributions across coalitions. More recently, NA$^2$Q [22] expands the joint value function via a Taylor-like decomposition to capture high-order interactions. However, such expansions scale exponentially with the number of agents, leading to substantial computational and interpretability challenges. These limitations highlight the need for a more principled and scalable formulation that can compactly model high-order agent interactions without sacrificing transparency. To this end, we propose a novel approach that integrates continued fraction networks into the value decomposition framework. By leveraging the recursive structure of continued fractions, our method enables compact and interpretable representations of arbitrary-order interactions while maintaining linear complexity with respect to the number of agents. This formulation provides a powerful alternative to polynomial expansion-based methods, offering both expressive capacity and interpretability in large-scale cooperative MARL settings.

# B Proof

## B.1 Objective Functions for Variational Information Bottlenecks

Considering the Markov chain $\boldsymbol{u}^* \leftrightarrow \boldsymbol{h} \leftrightarrow \boldsymbol{m}$, which means the assistive information cannot depend directly on the $\boldsymbol{u}^*$. So we have $p(\boldsymbol{m} \mid \boldsymbol{h}, \boldsymbol{u}^*) = p(\boldsymbol{m} \mid \boldsymbol{h})$.

As in the IB, the objective can be written as:

$$J_{IB}(\boldsymbol{\phi}) = I(\boldsymbol{m}, \boldsymbol{u}^*; \boldsymbol{\phi}) - \beta I(\boldsymbol{m}, \boldsymbol{h}; \boldsymbol{\phi}). \tag{15}$$

The $\beta$ is to realize the trade-off between a succinct representation and inferencing ability.

**Theorem 1** (Lower Bound for $I(\boldsymbol{m}, \boldsymbol{u}^*; \boldsymbol{\phi})$). *Let the representation $m_i$ be reparameterized as a random variable drawn from a multivariate Gaussian distribution $m_i \sim \mathcal{N}(f_m(h_i; \phi_m), \boldsymbol{I})$, where $f_m$ is an encoder parameterized by $\phi_m$, $h_i$ denotes the hidden state of agent $i$, and $\boldsymbol{I}$ is the identity covariance matrix. Then, the mutual information between the assistive information $\boldsymbol{m}$ and the optimal joint action $\boldsymbol{u}^*$ is lower-bounded as:*

$$I(\boldsymbol{m}, \boldsymbol{u}^*; \boldsymbol{\phi}) \geq \tfrac{1}{N} \sum_{i=1}^{N} \mathbb{E}_{\epsilon \sim p(\epsilon)} \left[ -\log q(u_i^* \mid f(h_i, \epsilon)) \right], \tag{16}$$

*where $q(u_i^* \mid m_i)$ is a variational distribution approximating the true posterior $p(u_i^* \mid m_i)$, and $m_i = f(h_i, \epsilon)$ denotes a deterministic function of $h_i$ and the Gaussian random variable $\epsilon$.*

*Proof.*

$$I(\boldsymbol{m}, \boldsymbol{u}^*; \boldsymbol{\phi}) = \int dm_i du_i^* p(m_i, u_i^*) \log \frac{p(m_i, u_i^*)}{p(m_i) p(u_i^*)}$$

$$= \int dm_i du_i^* p(m_i, u_i^*) \log \frac{p(u_i^* \mid m_i)}{p(u_i^*)},$$

where $p(u_i^* \mid m_i)$ is fully defined by our encoder and Markov Chain as follows:

$$p(u_i^* \mid m_i) = \int dh_i p(h_i, u_i^* \mid m_i)$$

$$= \int dh_i p(u_i^* \mid h_i) p(h_i \mid m_i)$$

$$= \int dh_i \frac{p(u_i^* \mid h_i) p(m_i \mid h_i) p(h_i)}{p(m_i)}.$$

Since this is intractable in our case, let $q(u_i^* \mid m_i)$ be a variational approximation to $p(u_i^* \mid m_i)$, where this is our decoder which we will take to another neural network with its own set of parameters. Using the fact that Kullback Leibler divergence is always positive, we have

$$\mathrm{KL}[p(u_i^* \mid m_i), q(u_i^* \mid m_i)] \geq 0$$

$$\implies \int du_i^* p(u_i^* \mid m_i) \log p(u_i^* \mid m_i) \geq \int du_i^* p(u_i^* \mid m_i) \log q(u_i^* \mid m_i),$$

and hence

$$I(\boldsymbol{m}, \boldsymbol{u}^*; \boldsymbol{\phi}) \geq \int du_i^* dm_i p(u_i^*, m_i) \log \frac{q(u_i^* \mid m_i)}{p(u_i^*)}$$

$$= \int du_i^* dm_i p(u_i^*, m_i) \log q(u_i^* \mid m_i) - \int du_i^* p(u_i^*) \log p(u_i^*)$$

$$= \int du_i^* dm_i p(u_i^*, m_i) \log q(u_i^* \mid m_i) + H(u_i^*)$$

$$= \int dh_i du_i^* dm_i p(h_i) p(u_i^* \mid h_i) p(m_i \mid h_i) \log q(u_i^* \mid m_i) + H(u_i^*)$$

$$= \frac{1}{N} \sum_{i=1}^{N} \left[ \int dm_i p(m_i \mid \tau_i) \log q(u_i^* \mid m_i) \right] + H(u_i^*).$$

Notice that the entropy of our labels $H(u_i^*)$ is independent of our optimization procedure and so can be ignored. And as we can rewrite $p(m_i \mid h_i)dm_i = p(\epsilon)d\epsilon$, $m_i = f(h_i, \epsilon)$. So we have

$$I(\boldsymbol{m}, \boldsymbol{u}^*; \boldsymbol{\phi}) \geq \frac{1}{N} \sum_{i=1}^{N} \mathbb{E}_{\epsilon \sim p(\epsilon)} \left[ -\log q(u_i^* \mid f(h_i, \epsilon)) \right].$$

$\square$

**Theorem 2** (Upper Bound for $I(\boldsymbol{m}, \boldsymbol{h}; \boldsymbol{\phi})$)**.** *Let $\tilde{q}(\boldsymbol{m})$ denote a variational approximation of the marginal distribution $p(\boldsymbol{m})$. Then, the mutual information between the representation $\boldsymbol{m}$ and the hidden state $\boldsymbol{h}$ admits the following upper bound:*

$$I(\boldsymbol{m}, \boldsymbol{h}; \boldsymbol{\phi}) \leq \mathrm{KL}(p(\boldsymbol{m} \mid h_i) \,\|\, \tilde{q}(\boldsymbol{m})). \tag{17}$$

*Proof.*

$$I(\boldsymbol{m}, \boldsymbol{h}; \boldsymbol{\phi}) = \int dm_i dh_i p(h_i, m_i) \log \frac{p(m_i \mid h_i)}{p(m_i)}$$

$$= \int dm_i dh_i p(h_i, m_i) \log p(m_i \mid h_i) - \int dm_i p(m_i) \log p(m_i).$$

Let $\tilde{q}(m_i)$ be the variational approximation to the marginal distribution $p(m_i) = \int dh_i p(m_i \mid h_i)p(h_i)$. Since $\mathrm{KL}[p(m_i), \tilde{q}(m_i)] \geq 0 \implies \int dm_i p(m_i) \log p(m_i) \geq \int dm_i p(m_i) \log \tilde{q}(m_i)$, we have

$$I(\boldsymbol{m}, \boldsymbol{h}; \boldsymbol{\phi}) \leq \int dh_i dm_i p(h_i)p(m_i \mid h_i) \log \frac{p(m_i \mid h_i)}{\tilde{q}(m_i)}$$

$$= \frac{1}{N} \sum_{i=1}^{N} \left[ p(m_i \mid h_i) \log \frac{p(m_i \mid h_i)}{\tilde{q}(m_i)} \right]$$

$$= \mathrm{KL}\left[ p(\boldsymbol{m} \mid h_i), \tilde{q}(\boldsymbol{m}) \right].$$

$\square$

Combining *Theorem 1* and *Theorem 2*, we have the objective functions for variational information bottlenecks, which is to minimize

$$\mathcal{L}_{VIB} = \frac{1}{N} \sum_{i=1}^{N} \mathbb{E}_{\epsilon \sim p(\epsilon)} \left[ -\log q\left(u_i^* \mid f(h_i, \epsilon)\right) \right] + \beta \mathrm{KL}\left[ p\left(\boldsymbol{m} \mid h_i\right), \tilde{q}(\boldsymbol{m}) \right]. \tag{18}$$

## B.2  Correspondence between Continued Fraction Depth and the Order of Agent Interactions

In this section, we establish a one-to-one correspondence between the depth $d$ of the continued fraction network and the order of agent interactions. This property allows the $d$-th order continued fraction to accurately represent the $d$-th order approximation of the agent's behavior.

Specifically, a continued fraction network of depth $d$, $\frac{1}{w_1 Q +} \frac{1}{w_2 Q +} \cdots \frac{1}{w_d Q}$ can be reformulated as $f(\boldsymbol{Q}) = T_d(\boldsymbol{Q}) + \mathcal{O}_{d+1}(\boldsymbol{Q})$, where $T_n(\boldsymbol{Q})$ is a degree-$d$ polynomial of $\boldsymbol{Q}$, and $\mathcal{O}_{d+1}(\boldsymbol{Q})$ denotes terms of order $d+1$ or higher in $\boldsymbol{Q}$.

By setting $\boldsymbol{z} = \frac{1}{\boldsymbol{Q}}$, the continued fraction $\frac{1}{w_1 Q +} \frac{1}{w_2 Q +} \frac{1}{w_3 Q +} \cdots$ can be transformed into

$$\mathbf{K}(\boldsymbol{z}) = \frac{\boldsymbol{z}}{\boldsymbol{w_1}} \frac{\boldsymbol{z}}{\boldsymbol{w_2}} \frac{\boldsymbol{z}}{\boldsymbol{w_3} + \cdots}, \tag{19}$$

since these approximants are arranged along the "staircase diagonals" of the Padé table.

**Theorem 3.** *For the $d$-th order truncation of the continued fraction $R_k(\boldsymbol{z}) = \frac{A_k(\boldsymbol{z})}{B_k(\boldsymbol{z})}$, the following holds:*

$$p_d = \left\lfloor \frac{d+1}{2} \right\rfloor, \quad q_d = \left\lfloor \frac{d}{2} \right\rfloor, \tag{20}$$

*where $p_d = \deg(A_d), \quad q_d = \deg(B_d)$.*

*Proof.* The $k$-th order asymptotic function $\frac{A_k(z)}{B_k(z)}$ satisfies the recursive relations:

$$\begin{cases} A_k(z) = w_k A_{k-1}(z) + z A_{k-2}(z) \\ B_k(z) = w_k B_{k-1}(z) + z B_{k-2}(z) \end{cases},$$

with

$$\begin{bmatrix} A_{-1} & A_0 \\ B_{-1} & B_0 \end{bmatrix} = \begin{bmatrix} 1 & 0 \\ 0 & 1 \end{bmatrix}.$$

Assume that for all $k \leq n$, the following holds:

$$\deg(A_k) = \left\lfloor \frac{k+1}{2} \right\rfloor, \quad \deg(B_k) = \left\lfloor \frac{k}{2} \right\rfloor.$$

**Base Cases:**

$n = 1$:

$$A_1(z) = w_1 A_0(z) + z A_{-1}(z) = z \Rightarrow \deg(A_1) = 1, \quad \left\lfloor \frac{1+1}{2} \right\rfloor = 1.$$

$$B_1(z) = w_1 B_0(z) + z B_{-1}(z) = w_1 \Rightarrow \deg(B_1) = 0, \quad \left\lfloor \frac{1}{2} \right\rfloor = 0.$$

$n = 2$:

$$A_2(z) = w_2 A_1(z) + z A_0(z) = w_2 z \Rightarrow \deg(A_2) = 1, \quad \left\lfloor \frac{3}{2} \right\rfloor = 1.$$

$$B_2(z) = w_2 B_1(z) + z B_0(z) = w_1 w_2 + z \Rightarrow \deg(B_2) = 1, \quad \left\lfloor \frac{2}{2} \right\rfloor = 1.$$

Hence, the base cases hold.

Then assume the statements hold for $k = n - 1$ and $k = n - 2$, and prove that they also hold for $k = n$.

**Degree of the Numerator $A_n(z)$:**

From the recurrence:

$$A_n(z) = w_n A_{n-1}(z) + z A_{n-2}(z),$$

and by the induction hypothesis:

$$\deg(A_{n-1}) = \left\lfloor \frac{n}{2} \right\rfloor, \quad \deg(A_{n-2}) = \left\lfloor \frac{n-1}{2} \right\rfloor.$$

Then:

$$\deg(w_n A_{n-1}) = \left\lfloor \frac{n}{2} \right\rfloor, \quad \deg(z A_{n-2}) = 1 + \left\lfloor \frac{n-1}{2} \right\rfloor.$$

Case 1: $n$ even, let $n = 2m$:

$$\left\lfloor \frac{n}{2} \right\rfloor = m, \quad \left\lfloor \frac{n-1}{2} \right\rfloor = m - 1.$$

Then

$$\deg(w_n A_{n-1}) = m, \quad \deg(z A_{n-2}) = 1 + (m - 1) = m \Rightarrow \deg(A_n) = m = \left\lfloor \frac{n+1}{2} \right\rfloor.$$

Case 2: $n$ odd, let $n = 2m + 1$:

$$\left\lfloor \frac{n}{2} \right\rfloor = m, \quad \left\lfloor \frac{n-1}{2} \right\rfloor = m.$$

Then

$$\deg(w_n A_{n-1}) = m, \quad \deg(z A_{n-2}) = 1 + m = m + 1 \Rightarrow \deg(A_n) = m + 1 = \left\lfloor \frac{n+1}{2} \right\rfloor.$$

**Degree of the Denominator $B_n(z)$:**

From the recurrence:
$$B_n(z) = w_n B_{n-1}(z) + z B_{n-2}(z),$$
and using the induction hypothesis:
$$\deg(B_{n-1}) = \left\lfloor \frac{n-1}{2} \right\rfloor, \quad \deg(B_{n-2}) = \left\lfloor \frac{n-2}{2} \right\rfloor.$$

Then:
$$\deg(w_n B_{n-1}) = \left\lfloor \frac{n-1}{2} \right\rfloor, \quad \deg(z B_{n-2}) = 1 + \left\lfloor \frac{n-2}{2} \right\rfloor.$$

Case 1: $n = 2m$ (even):
$$\left\lfloor \frac{n-1}{2} \right\rfloor = m - 1, \quad \left\lfloor \frac{n-2}{2} \right\rfloor = m - 1.$$

Then
$$\deg(w_n B_{n-1}) = m - 1, \quad \deg(z B_{n-2}) = m \Rightarrow \deg(B_n) = m = \left\lfloor \frac{n}{2} \right\rfloor.$$

Case 2: $n = 2m + 1$ (odd):
$$\left\lfloor \frac{n-1}{2} \right\rfloor = m, \quad \left\lfloor \frac{n-2}{2} \right\rfloor = m - 1.$$

Then
$$\deg(w_n B_{n-1}) = m, \quad \deg(z B_{n-2}) = m \Rightarrow \deg(B_n) = m = \left\lfloor \frac{n}{2} \right\rfloor.$$

By mathematical induction, we conclude that:
$$\deg(A_n) = \left\lfloor \frac{n+1}{2} \right\rfloor, \quad \deg(B_n) = \left\lfloor \frac{n}{2} \right\rfloor.$$

Therefore, when the truncation order is $n = d$, we have
$$p_d = \left\lfloor \frac{d+1}{2} \right\rfloor, \quad q_d = \left\lfloor \frac{d}{2} \right\rfloor.$$

$\square$

**Theorem 4.** *The $d$-th order truncation of the continued fraction $R_d(z) = \frac{A_d(z)}{B_d(z)}$ naturally satisfies the conditions for a Padé approximant, specifically:*
$$f(z) - R_d(z) = \mathcal{O}(z^{p_d + q_d + 1}), \tag{21}$$
*which means that its Taylor expansion coincides with the first $d$ terms of the original function $f(z)$.*

*Proof.*

**Definition 1** (Padé Approximant [25, 26])**.** *Let $C(z) = \sum_{k=0}^{\infty} c_k z^k$ be a formal power series in the variable $z$, then the Padé approximant of order $[L/M]$ is a rational function of the form:*
$$R_{L,M}(z) = [A_{L,M}(z)]/[B_{L,M}(z)], \tag{22}$$
*where $A_{L,M}(z)$ and $B_{L,M}(z)$ are polynomials of degrees at most $L$ and $M$, respectively, chosen such that*
$$B_{L,M}(z)C(z) - A_{L,M}(z) = \mathcal{O}(z^{L+M+1}), \tag{23}$$
*where notation $\mathcal{O}(z^k)$ denotes some power series of the form $\sum_{n=k}^{\infty} \tilde{c}_n z^n$. This approximation minimizes the difference between the rational function and the power series up to the order $L + M$.*

Since $R_d(z) = \frac{A_d(z)}{B_d(z)}$, we have
$$f(z) - \frac{A_d(z)}{B_d(z)} = \mathcal{O}(z^{p_d + q_d + 1}), \tag{24}$$

which implies
$$f(z)B_d(z) - A_d(z) = \mathcal{O}\left(z^{p_d + q_d + 1}\right) B_d(z) = \mathcal{O}\left(z^{p_d + q_d + 1}\right). \tag{25}$$

**Lemma 1.** *For all $k \geq 0$, there exists a polynomial $S_k(z)$ such that:*

$$f(z)B_k(z) - A_k(z) = (-1)^k z^{k+1} S_k(z), \qquad (26)$$

*and the constant term $S_k(0) \neq 0$.*

*Proof.* We can prove this lemma by mathematical induction.

**Base Case:**

$k = 0$:

$$f(z)B_0(z) - A_0(z) = f(z) \cdot 1 - 0 = f(z).$$

By the definition of continued fractions, $f(z) = \frac{z}{a_1 + \cdots}$, so:

$$f(z) = z \cdot (\text{analytic function}) = z S_0(z), \quad S_0(0) = \frac{1}{a_1} \neq 0.$$

**Inductive Hypothesis ($k - 1$ and $k - 2$ hold):**

$$f(z)B_{k-1}(z) - A_{k-1}(z) = (-1)^{k-1} z^k S_{k-1}(z)$$
$$f(z)B_{k-2}(z) - A_{k-2}(z) = (-1)^{k-2} z^{k-1} S_{k-2}(z)$$

**for $k$:**

Substituting the recurrence relations:

$$
\begin{aligned}
f(z)B_k(z) - A_k(z) &= f(z)\left(w_k B_{k-1}(z) + z B_{k-2}(z)\right) - \left(w_k A_{k-1}(z) + z A_{k-2}(z)\right) \\
&= w_k\left(f(z)B_{k-1}(z) - A_{k-1}(z)\right) + z\left(f(z)B_{k-2}(z) - A_{k-2}(z)\right) \\
&= w_k(-1)^{k-1} z^k S_{k-1}(z) + z(-1)^{k-2} z^{k-1} S_{k-2}(z) \\
&= (-1)^k z^k \left(-w_k S_{k-1}(z) + S_{k-2}(z)\right) \\
&= (-1)^k z^{k+1} S_k(z),
\end{aligned}
$$

where $S_k(z)$ is the polynomial obtained from $-w_k S_{k-1}(z) + S_{k-2}(z)$, divided by $z$. $\qquad \square$

From the lemma, we have:

$$f(z) - \frac{A_d(z)}{B_d(z)} = \frac{f(z)B_d(z) - A_d(z)}{B_d(z)} = \frac{(-1)^d z^{d+1} S_d(z)}{B_d(z)}.$$

Since $B_d(0) = w_1 w_2 \cdots w_d \neq 0$ (assuming $w_k \neq 0$) and $S_d(0) \neq 0$, it follows that:

$$f(z) - R_d(z) = \mathcal{O}(z^{d+1}).$$

According to *Theorem 3*, the degree of the numerator $A_d(z)$ is $p_d = \left\lfloor \frac{d+1}{2} \right\rfloor$, and the degree of the denominator $B_d(z)$ is $q_d = \left\lfloor \frac{d}{2} \right\rfloor$. When $d$ is odd, we have $p_d = q_d = \frac{d}{2}$; when $d$ is even, $p_d = \frac{d+1}{2}$, $q_d = \frac{d-1}{2}$. In both cases, it follows that $p_d + q_d = d$. Therefore,

$$f(z) - R_d(z) = \mathcal{O}\left(z^{d+1}\right) = \mathcal{O}\left(z^{p_d + q_d + 1}\right),$$

which satisfies the condition of a Padé approximant. $\qquad \square$

In conclusion, the depth-$d$ continued fraction network represents the $d$-th order truncation of the continued fraction:

$$\frac{1}{w_1 Q +} \frac{1}{w_2 Q +} \cdots \frac{1}{w_d Q},$$

which forms a $[p_d, q_d]$-Padé approximant with $p_d + q_d = d$. This enables accurate representation of the first $d$-th order interactions among agents.

# C   Experimental Details

## C.1   Algorithmic Description

---

**Algorithm 1** Continued Fraction Q-Learning

---

1: Initialize environment, agent network $Q_i(\tau_i, u_i; \theta)$, target network $Q_i\left(\tau_i', u_i'; \hat{\theta}\right)$, mixing net-
   work $Q_{tot}$, and VIB module $G_\phi(E_{\phi_1}, D_{\phi_2})$
2: Initialize replay buffer $\mathcal{D}$
3: **repeat**
4:     Obtain the initial global state $s^0$
5:     **for** $t = 0$ to $T - 1$ **do**
6:         For each agent i, get action-observation history $\tau_i^t$
7:         Calculate individual value function $Q_i$
8:         Get the hidden state $h_i^t$
9:         Select action $u_i^t$ via value function with probability $\epsilon$ exploration
10:        Execute joint action $\boldsymbol{u^t}$, receive reward $r^t$, next state $s^{t+1}$
11:    **end for**
12:    Store the episode trajectory in $\mathcal{D}$
13:    Sample a mini-batch $\mathcal{B}$ of size $b$ from $\mathcal{D}$
14:    **for** $t = 0$ to $T - 1$ **do**
15:        Calculate $\mu, \sigma = E_{\phi_1}(h_i^t)$
16:        Generate assistive information $\boldsymbol{m}$
17:        Get the attention weight $\alpha_k$ by the intervention function in Eq. 12
18:        Calculate the joint value function $Q_{tot}$
19:    **end for**
20:    Calculate loss $\mathcal{L}(\theta) = \mathcal{L}_{Q_{tot}} + \mathcal{L}_{VIB}$ via Eq. 11 and Eq. 14.
21:    Update $\phi$ and $\theta$ by minimizing the above loss
22:    Periodically update $\hat{\theta} \leftarrow \theta$
23: **until** $Q_i(\tau_i, u_i; \theta)$ converges or maximum steps reached

---

## C.2   LBF Description and Hyperparameters Settings

Level-Based Foraging (LBF) [31] is a mixed cooperative-competitive MARL benchmark, where each agent navigates a $10 \times 10$ grid world. Agents and food items are randomly placed in a 2D grid, and each one is assigned a level. A food item can only be collected when the combined levels of all participating agents equal or exceed its level. The environment induces a spectrum of collaborative behaviors through its level-dependent reward structure: while low-level food items permit independent collection, higher-level resources necessitate coalition formation. Furthermore, we set the penalty reward for movement to $-0.002$, and the detailed hyperparameter settings of LBF are shown in Table 1.

Table 1: The configurations of LBF.

| Hyperparameter | Value |
|---|---|
| Max Episode Length | 50 |
| Batch Size | 32 |
| Test Interval | 10000 |
| Test Episodes | 32 |
| Replay Batch Size | 5000 |
| Discount Factor | 0.99 |
| Start Exploration Rate | 1.0 |
| End Exploration Rate | 0.05 |
| Anneal Steps | 50000 |
| Steps | 1M |
| Target Update Interval | 200 |

**Observation Space.** Each agent observes a $2 \times 2$ square grid centered on its own position. Within this range, the agent receives a structured array containing the $(x, y)$ coordinates and levels of all visible food items and other agents. This observation provides both spatial and attribute-level information to support localized decision-making.

**Action Space.** The discrete action space for each agent consists of none, move [direction], and load food. Each agent only moves into one unoccupied grid. If multiple agents attempt to move into the same grid, collisions are resolved by canceling the conflicting moves, leaving the agents in their original positions.

**Rewards.** This reward depends on the food's level, which is distributed among the participating agents in proportion based on their levels. The rewards are normalized to maintain a unit sum across all agents. This design ensures contribution-based fairness in reward distribution while enhancing cooperative efficiency among agents.

## C.3 StarCraft II Description and Hyperparameters Settings

All implementations of algorithms are conducted on Star-Craft II unit micro-management tasks (SC2.4.10). We evaluate performance in combat scenarios where enemy units are controlled by the built-in AI with the *difficulty=7* setting, and each allied unit is controlled by the decentralized agents with reinforcement learning algorithms. During battles, the agents seek to maximize the damage dealt to enemy units while minimizing damage received, requiring the coordination of diverse tactical skills. We assess our method across a variety of challenging scenarios that differ in terms of symmetry, agent composition, and unit count (as shown in Table 3). For clarity, we also outline the core settings of the StarCraft Multi-Agent Challenge (SMAC) [32], including observation, state, action, and reward configurations. The detailed hyperparameter settings of SMAC are shown in Table 2.

Table 2: The configurations of SMAC.

| Hyperparameter | Value |
|---|---|
| Difficulty | 7 |
| Batch Size | 32 |
| Test Interval | 10000 |
| Test Episodes | 32 |
| Replay Batch Size | 5000 |
| Discount Factor | 0.99 |
| Start Exploration Rate | 1.0 |
| End Exploration Rate | 0.05 |
| Target Update Interval | 200 |
| Optimizer | RMSprop |
| Learning Rate | 0.0005 |

**Observations and States.** At each time step, each agent receives a local observation of units within its field of view. The observation includes the following features for both allied and enemy units: distance, relative X and Y positions, health, shield, and unit type. Note that the agents can only observe the others if they are alive and within their line of sight range, which is set to 9. When a unit (ally or enemy) becomes invisible or is eliminated, its feature vector is reset to all zeros, indicating either death or being outside the field of view. The global state is only available during centralized training, which contains information about all units on the map. Finally, all features, including the global state and the observation of the agent, are normalized by their maximum values.

**Action Space.** Each unit takes an action from the discrete action set: no-op, stop, move [direction], and attack [enemy id]. Agents are allowed to move with a fixed movement amount in four directions: north, south, east, and west, where the unit is allowed to take the attack [enemy id] action only when the enemy is within its shooting range.

**Rewards.** The target goal is to maximize the win rate for each battle scenario. At each time step, the agents receive a shaped reward based on the hit-point damage dealt and enemy units killed, as well as a special bonus for winning the battle. Additionally, agents obtain a 10 positive bonus after killing each enemy and a 200 bonus when killing all enemies, which is consistent with the default reward function of the SMAC.

Table 3: The StarCraft Multi-Agent Challenge benchmark.

| Map | Ally Units | Enemy Units | Difficulty | Steps | Anneal Steps | $d$ |
|---|---|---|---|---|---|---|
| 2s3z | 2 Stalkers, 3 Zealots | 2 Stalkers, 3 Zealots | Eazy | 1.5M | 50000 | 2 |
| 2c_vs_64zg | 2 Colossus | 64 Zerglings | Hard | 2M | 50000 | 2 |
| 3s_vs_5z | 3 Stalkers | 5 Zealots | Hard | 2M | 50000 | 2 |
| 5m_vs_6m | 5 Marines | 6 Marines | Hard | 2M | 50000 | 4 |
| 3s5z_vs_3s6z | 3 Stalkers, 5 Zealots | 3 Stalkers, 6 Zealots | Super Hard | 5M | 200000 | 6 |
| 6h_vs_8z | 6 Hydralisks | 8 Zealots | Super Hard | 5M | 200000 | 6 |

## C.4 SMACv2 Description and Hyperparameters Settings

SMACv2 [33] is an enhanced benchmark for cooperative multi-agent reinforcement learning built on top of StarCraft II. It preserves the original SMAC API while introducing three procedural innovations to increase scenario diversity and challenge contemporary MARL algorithms: randomising start positions, randomising unit types, and changing the unit sight and attack ranges.

**Randomized Start Positions.** Allied and enemy units are spawned either in a "surround" configuration, where enemies encircle the allies, or via a "reflect" scheme that mirrors allied positions across the map center. This ensures that agents cannot overfit to fixed spawn patterns.

**Randomized Unit Types.** Each battle can feature mixed unit compositions rather than uniform rosters. For Terran, Protoss, and Zerg, three unit types are sampled with configurable probabilities through the team_gen distribution (as shown in Table 4), promoting adaptable strategies under varied team makeups.

**Unit Sight and Attack Ranges.** Unit vision cones and attack radii are aligned with their true in-game values, increasing realism and preventing agents from exploiting the simplified ranges used in SMAC.

Table 4: The configurations of SMACv2.

| Race | Unit | probability |
|---|---|---|
| | Stalker | 0.45 |
| Protoss | Zealot | 0.45 |
| | Colossus | 0.1 |
| | Marine | 0.45 |
| Terran | Marauder | 0.45 |
| | Medivac | 0.1 |
| | Zergling | 0.45 |
| Zerg | Hydralisk | 0.45 |
| | Baneling | 0.1 |

## C.5 Implementation Details

We compare our method against nine value-based baselines, including VDN [21], QMIX [11], QPLEX [34], Centrally-Weighted QMIX (CW-QMIX) [35], CDS [36], SHAQ [37], GoMARL [38], ReBorn [39], and NA$^2$Q [22]. To ensure fairness, we implement all experiments within the PyMARL framework [2]. All hyperparameters of baselines are set identically to our method to compare algorithms fairly. Please refer to PyMARL's open-source implementation for further training details and fair comparison settings. The depth $d$ of CFN is determined based on the scale of agents and the complexity of each task.

All scenarios are trained on a system equipped with an NVIDIA RTX 3080TI GPU and an Intel i9-12900k CPU, with training time ranging from 1 to 16 hours per scenario, depending on the task complexity and episode length.

---

[2]The source code of implementations is from `https://github.com/oxwhirl/pymarl`.

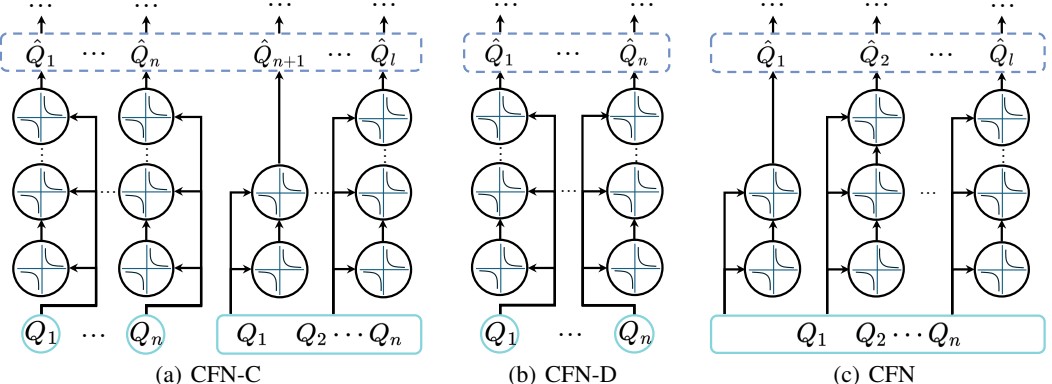

Figure 10: Three variants of the CFN architecture. (a) CFN-C integrates both single-feature ladders, where each ladder processes a single input dimension $Q_i$, and full ladders, which take the complete set of individual values $\boldsymbol{Q}$ as input. (b) CFN-D utilizes only the single-feature ladders. (c) CFN employs only the full ladders with increasing depth.

## C.6 Detailed Description of CFN Structure

As illustrated in *Fig.10*, the CFN framework includes two structural variants in addition to the main architecture. *Fig. 10(a)* presents CFN-C, a composite architecture inspired by CoFrNet [18], which combines two types of ladders: single-feature ladders, each processing an individual agent utility $Q_i$, and full-input ladders, which receive the complete utility vector $\boldsymbol{Q}$ at every layer. Each ladder yields a partial joint value $\hat{Q}_k$, and the aggregation of all ladders constitutes the final joint Q-value.

The number of single-feature ladders equals the number of agents, enabling additive modeling of individual effects. In contrast, full-input ladders are deeper and designed to capture complex joint dependencies among agents by recursively combining all inputs, thereby facilitating high-order interaction modeling.

*Fig. 10(b)* and *Fig. 10(c)* depict two simplified variants: 1) CFN-D, which retains only the single-feature ladders, thereby modeling additive effects with strong transparency [47] but lacking the capacity to express inter-agent interactions; 2) CFN, which retains only the full-input ladders, striking a balance between modeling power and computational efficiency.

In our QCoFr algorithm, we adopt the CFN structure with full-input ladders as the default architecture. Compared to CFN-C, this version significantly reduces parameter overhead while preserving the ability to capture arbitrary-order interactions. We further include CFN-D as an ablation baseline to isolate the contribution of high-order modeling: while CFN-D offers interpretability due to its decomposable additive form, its inability to encode dependencies across agents limits its expressiveness in cooperative settings.

Finally, a key advantage of the CFN structure is its linear scalability: the number of parameters grows as $\mathcal{O}(n)$ with the number of agents, making it particularly suited to large-scale MARL scenarios where modeling expressive joint behavior is critical without incurring prohibitive computational costs.

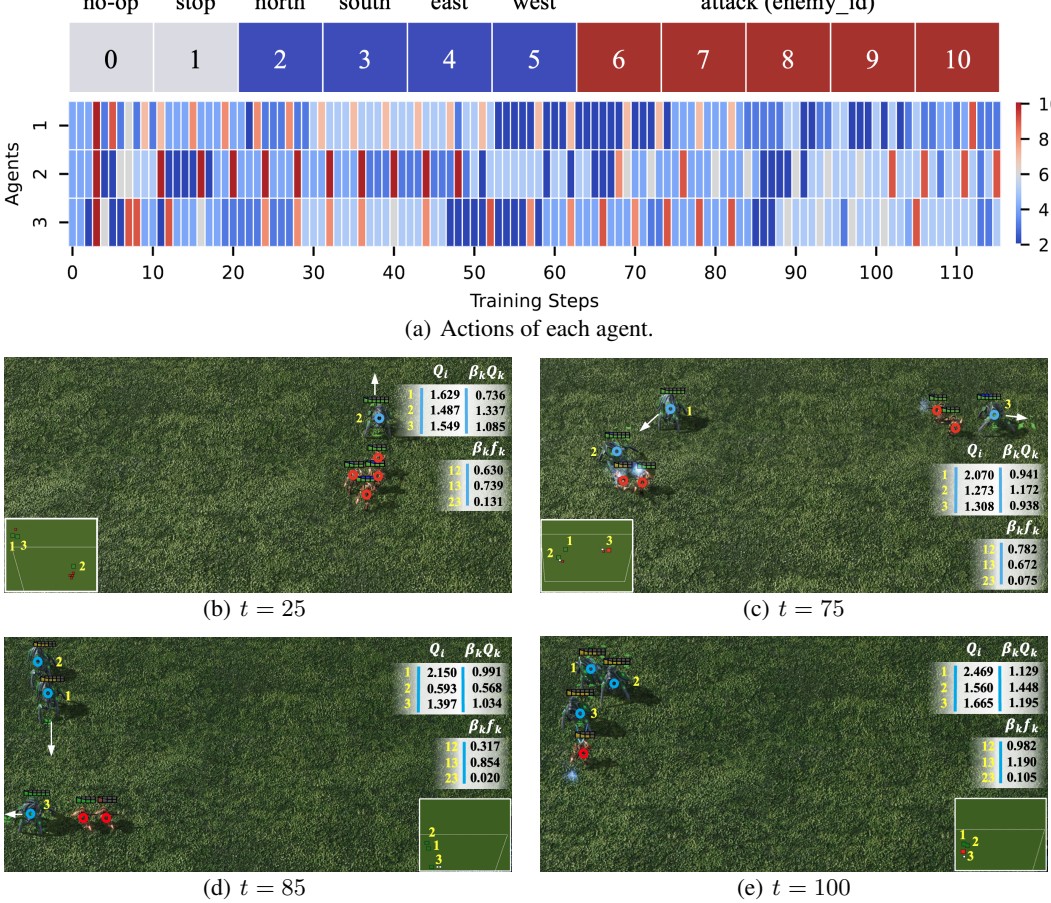

Figure 11: Visualization of evaluation results for QCoFr on *3s_vs_5z* map. Agents demonstrate a coordinated kite-and-focus-fire strategy: agent 2 initially kites four enemies alone, while agent 1 and agent 3 eliminate another. Agent 3 then draws away two of the remaining enemies, enabling agent 1 and agent 2 to dispatch the others. Finally, all agents regroup to defeat the last enemies.

## D  Extended Interpretability Analysis

*Fig. 11* illustrates the interpretability of QCoFr on *3s_vs_5z* scenario. At the beginning of the episode, agent 2 independently kites four enemies, creating a numerical advantage that enables agent 1 and agent 3 to quickly eliminate an isolated opponent. As a result, agent 2 receives the highest individual contribution score (1.337), while the strongest pairwise contribution is observed between agents 1 and 3 due to their effective coordination. As the engagement progresses, agent 3 draws two enemies away, allowing agent 1 and agent 2 to jointly take down the remaining targets. During this phase, the coalition contribution of agents 1 and 2 increases, and agent 3's individual contribution also rises as it delays the enemy. After these enemies are defeated, all three agents regroup to focus fire on the remaining units, resulting in a more balanced distribution of credit across agents. This case study demonstrates that the agents have learned a kite-and-focus-fire strategy. The alignment between observed behaviors and quantitative contribution values confirms the interpretability of QCoFr, which faithfully attributes both individual and coalition-level contributions with high-order interactions in executing complex cooperative tactics.

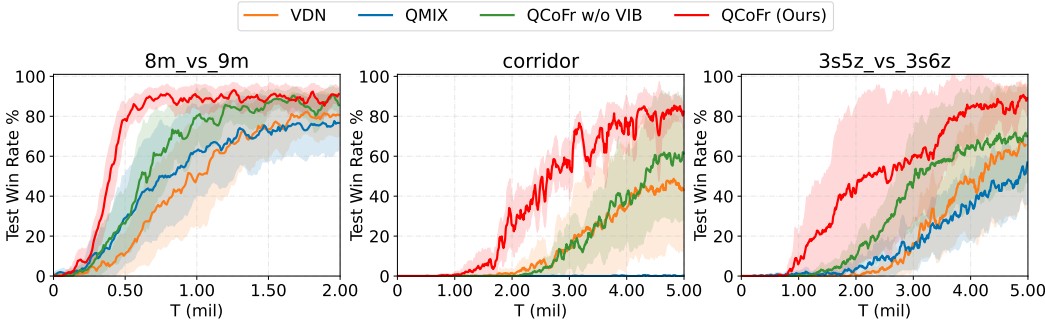

Figure 12: Performance with and without VIB on three extra scenarios of the SMAC benchmark.

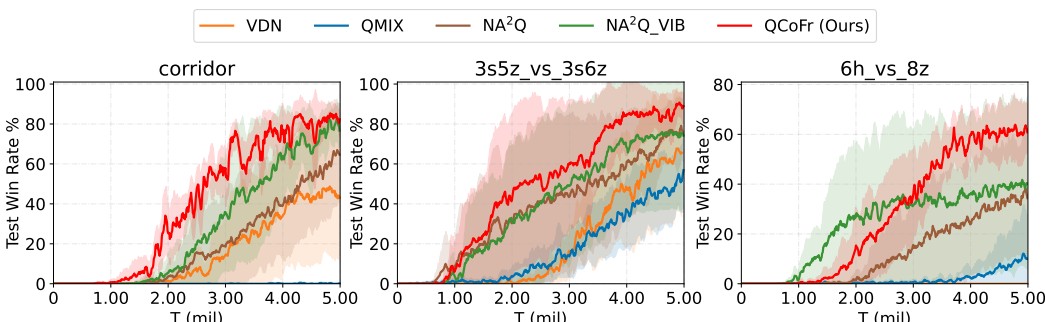

Figure 13: Performance comparison of NA$^2$Q with the VIB module and our method.

# E Additional Experiments on SMAC

## E.1 Additional Ablation Experiments

**The Role of the VIB Module.** We ablate the VIB component on three additional SMAC scenarios (*Fig. 12*), comparing QCoFr with and without VIB under identical settings. With VIB, QCoFr consistently accelerates early learning and achieves higher test win rates, confirming that task-relevant assistive information improves credit assignment and coordination.

**The Role of CFN.** Since NA$^2$Q struggles to model higher-order interactions, we equip it with the same VIB module and evaluate on three super-hard SMAC maps, comparing against QCoFr (*Fig. 13*). This isolates the effect of interaction modeling from that of assistive information. While NA$^2$Q+VIB outperforms the original NA$^2$Q, a clear gap remains to QCoFr. The results indicate that, on complex tasks, explicitly modeling higher-order dependencies enables more refined cooperative strategies, which highlights the effectiveness of the CFN module beyond what VIB alone can provide.

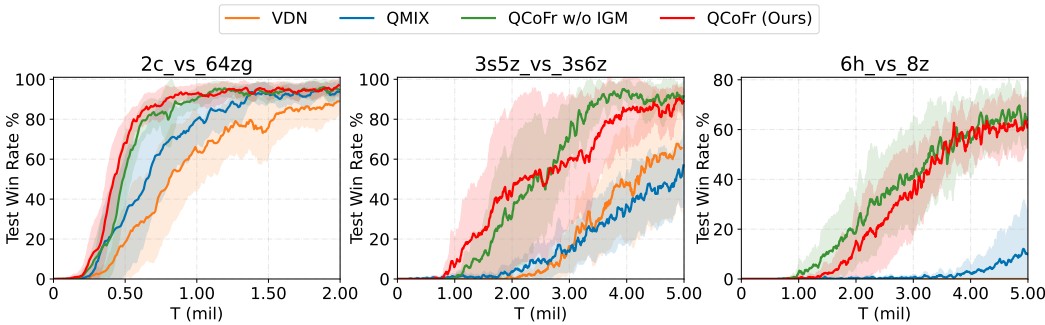

Figure 14: Performance comparison of QCoFr with and without IGM constraint.

## E.2 Discussion on the IGM Constraint

To isolate the effect of the continued-fraction mixing paradigm from non-monotonic joint-action search, we enforce the Individual-Global-Max (IGM) constraint in our framework. Notably, the universal approximation theorem applies to any linear combination of continued fractions and does not require non-negative weights [18], suggesting that the approach can be extended to non-IGM mixers. Integrating CFN with fully IGM-free methods such as DAVE [48] is therefore a natural direction. DAVE emphasizes that most value decomposition methods operate under IGM, which couples the optimal joint action with the optimal individual actions. Relaxing this constraint requires agents to explicitly search for the globally optimal joint action at execution time, often via an auxiliary network. To probe this possibility, we conduct experiments under relaxed IGM assumptions on three SMAC scenarios. As shown in *Fig. 14*, QCoFr achieves comparable or even slightly improved performance without IGM, indicating that our architecture can still recover high-quality joint actions.

