# OpenReview forum: "High-order Interactions Modeling for Interpretable Multi-Agent Q-Learning"
_NeurIPS.cc/2025/Conference — NeurIPS 2025 poster_

### Official Review · Reviewer_WW9h · 2025-06-03

**Clarity:** 3
**Significance:** 2
**Originality:** 2
**Rating:** 4
**Confidence:** 3

**Summary:**

Th paper introduce a novel multi-agent Q learning algorithm, named QCoFr. QCoFr models arbitrary-order interactions among agents using a Continued Fraction Network (CFN), which scales linearly with the number of agents and enables interpretability via rational approximation theory. The model incorporates a Variational Information Bottleneck (VIB) to distill environment-relevant latent information to aid credit assignment. Empirical results on LBF, SMAC, and SMACv2 benchmarks demonstrate competitive performance and improved interpretability over strong baselines.

**Questions:**

1. Why the VIB inputs the hidden state instead of the global sate?
2. Adaptive order could be explored.
3. Experiment on other multi-agent benchmark with more agent numbers could be added to explore the relation with interaction order and agent number.

**Ethical Concerns:**

["NO or VERY MINOR ethics concerns only"]

**Final Justification:**

All my concern is addressed . So I tend to accept.

**Limitations:**

Yes.

**Paper Formatting Concerns:**

nil

**Quality:**

3

**Strengths And Weaknesses:**

Strenghts:

1. The algorithm's performance is excellent on SMAC baselines.
2. The paper provides rigorous proofs around approximation guarantees, universal approximation, and information bottleneck bounds.
3. The use of continued fractions to represent high-order agent interactions is original

Weakness:
1. The VIB is introduced to address the causal confusion from global state. However, the VIB inputs the hidden state instead of the global sate.
2.  The CFN depth is fixed for per task. Adaptive mechanisms based on interaction complexity could be explored.
3. The figure could be make clear to illustrate the computation process for Q.

---

> ### Author Rebuttal · Authors · 2025-07-31
>
> Thank you for reviewing our paper and your constructive feedback. We appreciate the time and effort you spent on our paper and the helpful comments you provided. Please find our itemized responses to your questions below.
>
> 1. **The VIB take the hidden state as input.**
>
>    **Response:** Regarding the value decomposition in cooperative MARL, credit assignment serves an important role to guide the learning of decentralized policies, which aims to deduce the contributions of individual agents from the overall success. Since each agent observes the global state partially, there exists a potential backdoor path between the global state $\boldsymbol{s}$ and $Q_{tot}$, specifically $\boldsymbol{s} \leftarrow E \rightarrow \tau \rightarrow Q_{tot}$. This backdoor path can lead to a spurious correlation that causes confounding bias for estimating credit assignment. Inspired by the previous works [R1, R2], an effective solution is to impose an intervention function on $\boldsymbol{s}$ identifying the local history $\tau_i$ in an unobservable environment $E$. To realize it, we consider the information bottleneck method to produce assistive information $m$ aiding value factorization together global state $\boldsymbol{s}$, which forms the Markov chain $\boldsymbol{h}-\boldsymbol{m}-\boldsymbol{u^\*}$ during encoding. To learn an encoding that is maximally informative about the optimal action selection $\boldsymbol{u^\*}$, we regard the internal representation of the intermediate layer as stochastic encoding $\boldsymbol{m}$ of history $\boldsymbol{h}$. Thus, we write the objective as $J_{I B}(\boldsymbol{\phi})=I\left(\boldsymbol{m}, \boldsymbol{u}^\* ; \boldsymbol{\phi}\right)-\beta I(\boldsymbol{m}, \boldsymbol{h} ; \boldsymbol{\phi})$.
>
>
>     [R1] Jiahui Li, et.al., Deconfounded value decomposition for multi-agent reinforcement learning, ICML, 2022.
>
>     [R2] Zichuan Liu, et.al. NA$^2$Q : Neural attention additive model for interpretable multi-agent Q-learning. ICML, 2023.
>
>
> 2. **Adaptive order interactions.**
>
>     **Response:** We appreciate your insightful suggestion regarding adaptive adjustment of CFN depth. As noted in our limitations section, this is indeed a promising avenue for future work. Theoretically, increasing CFN depth allows the model to capture higher-order interactions among agents, improving expressiveness in tasks requiring coordination skills. However, unnecessarily modeling high-order interactions in simpler environments or with fewer agents may lead to overfitting or optimization instability.  To balance expressiveness with efficiency, one potential solution may be valuable for future exploration. Considering using adaptive computation, the CFN can be initialized with a base depth $d_i$ and iteratively explore neighboring depths within a radius $\delta_{d}$, halting exploration once marginal gains fall below a threshold. Another potential method is to integrate skip connections with learnable gating mechanisms, allowing the network to modulate depth in a differentiable and task-adaptive manner. While our current fixed-depth configuration already delivers strong empirical results, we agree that incorporating such adaptive depth control based on coordination complexity is a valuable future direction, and we will add this discussion to the revised manuscript.
>
> 3. **The computation process for $Q_{tot}$.**
>
>      **Response:** We thank the reviewer for pointing out the lack of clarity in the current figure illustrating the computation of the $Q_{tot}$. While the accompanying text describes the pipeline in detail, we acknowledge that the diagram could better reflect key components such as the recursive structure of the continued fraction network and the role of the auxiliary variable from the VIB module. In the new revision, we will update the figure to more explicitly highlight the sequential fraction aggregation process, clarify the data flow from the individual $Q_i$ values through CFN layers, and provide a clear explanation of the losses.  We appreciate your feedback again and are committed to improving the clarity of our presentation.
>
> 4. **Experiment on other multi-agent benchmark.**
>
>     **Response:**  Thank you for the valuable suggestion. To examine the relationship between interaction order and agent number, we conducted experiments on the **Multi-Agent COordination Challenge (MACO)** benchmark [R3], which comprises diverse multi-agent tasks with varying coordination demands and larger agent populations. We evaluated our method on three representative scenarios: **Pursuit** (12 agents), **Aloha** (15 agents), and **Gather** (10 agents). The results are summarized in Table 1, where $\pm$ denotes the standard deviation of the average performance over 5 trials, and the best performance in each scenario is highlighted in **bold**.
>
>    As summarized in **Table 1**, our method (QCoFr) significantly outperforms baselines such as VDN, QMIX, and NA$^2$Q with 2-order across all tasks, which should benefit from explicit modeling of higher-order interactions. As the orders increase, QCoFr (d=5) achieves better performance, which confirms that higher-order interactions are critical for coalitions with more agents to learn high-level cooperation skills. These results reinforce the theoretical motivation of our work: that increasing CFN depth improves performance in proportion to the coordination complexity and number of agents. We will include this discussion and full results in the revised manuscript.
>
>     ### Table 1. Performance comparison with baselines on the MACO benchmark.
>    | Method        |     pursuit       |       gather      |      aloha        |
>    | :------------:| :---------------: | :---------------: | :---------------: |
>    |               |  Test Prey Caught |   Test Win Rate   |  Test Transmitted |
>    | VDN           |   0.00$\pm$0.00   |   0.22$\pm$0.01   |   0.00$\pm$0.00   |
>    | QMIX          |   0.00$\pm$0.00   |   0.78$\pm$0.02   |   0.75$\pm$0.50   |
>    | NA$^2$Q       |   3.00$\pm$0.77   |   0.58$\pm$0.26   |   21.08$\pm$12.00  |
>    | QCoFr (d=3)  |     3.23$\pm$0.26  |    0.79$\pm$0.05  |   24.37$\pm$5.34   |
>    | QCoFr (d=5)  |   **3.75$\pm$0.15**   |   **0.90$\pm$0.01**   |   **28.50$\pm$7.13**   |
>
>
>     [R3] Tonghan Wang, et al., Context-Aware Sparse Deep Coordination Graphs. ICLR,2022.

---

> > ### Comment · Reviewer_WW9h · 2025-08-02
> > **Thanks for clarification.**
> >
> > Thank you for the responses. The additional experiments and clarifications have addressed my concerns. I don’t have further questions at this point.

---

> > > ### Author Response · Authors · 2025-08-02
> > >
> > > Thank you once again for taking out time to review our paper. We are very pleased to have addressed your concerns.

---

### Official Review · Reviewer_fozS · 2025-06-26

**Clarity:** 2
**Significance:** 2
**Originality:** 2
**Rating:** 4
**Confidence:** 2

**Summary:**

This paper presents a computational approach to modeling higher-order interactions in cooperative multi-agent settings. The paper formulates the fully cooperative multi-agent task as a decentralized partially observable Markov decision process. The goal is to learn an optimal joint policy that maximizes the joint value function. The authors employ value decomposition, which breaks down the global objective into local value functions while adhering to the individual-global-max (IGM) principle. These local Q-functions are combined to approximate the overall joint action-value function. The main contribution of the paper is the utilization of Continued Fraction Networks (CFNs) to capture higher-order interactions among the agents' Q-functions. The proposed model constructs the total Q-function by passing the individual Q-functions into a CFN to model higher-order interaction and then aggregating the outputs of the CFN using trained credit assignment functions. The paper's other contribution is the introduction of a Variational Information Bottleneck (VIB) instead of a variational autoencoder, as in previous literature. The VIB captures the most relevant information from history rather than using all of it to create the assistive information.

**Questions:**

A- According to our understanding, CFNs can suffer from instability with larger depths, this aligns with scalability issues of previous contributions. Can the authors expand on this point?

B- Interpretability is understandable given the power series connection. In terms of assigning credits, is there another interpretable way instead of using ReLUs?

C- What are possible considerations to move to fully decentralized approaches?

**Ethical Concerns:**

["NO or VERY MINOR ethics concerns only"]

**Final Justification:**

All my questions were answered as expected. I will keep my score as is.

**Limitations:**

Yes

**Quality:**

3

**Strengths And Weaknesses:**

Strengths:

1- The paper is well-written, but it requires some proofreading. The authors spent time comparing literature, providing figures, and writing detailed appendices.

2- The proposed method exhibits reasonable performance across various benchmark tasks.

3- The ablation studies are comprehensive, examining the effects of various models with different depths and structures, as well as studying the impact of the VIB module.

4-  Earlier studies on modeling higher-order interactions have relied on modeling only linear interactions, employing black-box implementations, or considering intractable models. In this case, the output of a fractional neural network can be represented as a power series, reflecting the interpretability of such models in characterizing interactions.

Weakness:

1- Authors refer to variables that are not clearly defined, such as $R_t$ and $Q_{tot}$. Although the intent is clear, such quantities should be defined clearly in the problem statement. If the issue is space, the authors can perhaps remove information such as Definition 1 from the main body (which was already defined in the appendix).



2- The structure of the paper follows a very similar structure to reference [21], which introduces NA$^2$Q. The intent in replacing the Neural Attention Additive model with CFN is clear in terms of tractability. Furthermore, this paper differs from [21] in the introduction of the VIB module, which extracts essential information from history. A few comments in this regard:







A- The performance of the presented approach is close to the performance of  [21],  except on tasks 3s_vs_5z, 3s5z_vs_3s6z, and 6h_vs_8z. However, as seen in the ablation test, VIB plays a significant role in shaping the performance of the current model in these tasks. For tasks 3s_vs_5z and 6h_vs_8z, the performance of the presented approach without VIB is comparable to that of [21]. To fully demonstrate the effect of modeling higher-order interactions, it is recommended to observe how [21] would perform with the addition of VIB to truly reflect the impact of modeling higher-order interactions using CFNs.







B- For the sake of completeness of comparison with [21], consider showing the performance on the SMAC benchmark for "Corridor" and "8m_vs_9m".


3- Local fixes are needed, some examples below:

- Line 45: "but also the interpretability" the sentence needs a verb such as provides



- Line 73: "For each coalition, it is formed. The usage of for is weird. Consider just saying, ",where each coalition is formed ..."


- Line 187, the sentence is weird.


- Line 30: consider removing the word "interactions."

---

> ### Author Rebuttal · Authors · 2025-07-31
>
> Thank you very much for your positive review! Below, we respond to your critical points and questions.
>
> 1. **Differences from NA$^2$Q.**
>
>    **Response:** Due to the limited space, please refer to the response to **Reviewer CAhf**.
>
> 2. **The role of CFN.**
>
>    **Response:** To fully demonstrate the effect of modeling higher-order interactions, we conducted experiments with NA$^2$Q combined with the VIB module on three super-hard maps, with the results summarized in Table 1. It is clear that the performance of NA$^2$Q with the VIB module is improved compared to the original NA$^2$Q, but there remains a noticeable gap between it and our method. This indicates that, in more complex tasks, modeling higher-order interactions allows agents to learn refined cooperation strategies, thereby demonstrating the effectiveness of the CFN module.
>
>     ### Table 1. Performance comparison of NA$^2$Q with the VIB module and our method.
>     | Method        |     corridor    |     3s5z_vs_3s6z  |      6h_vs_8z     |
>     | :------------:| :---------------: | :---------------: | :---------------: |
>     | VDN           |   0.46$\pm$0.36   |   0.64$\pm$0.33   |   0.00$\pm$0.00   |
>     | QMIX          |   0.01$\pm$0.01   |   0.61$\pm$0.19   |   0.11$\pm$0.19   |
>     | NA$^2$Q       |   0.63$\pm$0.20   |   0.73$\pm$0.14   |   0.30$\pm$0.24   |
>     |  NA$^2$Q_VIB  |   0.77$\pm$0.06   |   0.73$\pm$0.36   |   0.39$\pm$0.34   |
>     | QCoFr (Ours)  |   **0.79$\pm$0.03**   |   **0.87$\pm$0.07**   |   **0.58$\pm$0.12**   |
>
>
>
> 4. **Performance on the SMAC benchmark for "Corridor" and "8m_vs_9m" compared with NA$^2$Q.**
>
>    **Response:** The performance comparison between our method and NA$^2$Q on the Corridor and 8m\_vs\_9m maps is shown in Table 2. In both maps, our method outperforms NA$^2$Q in terms of both convergence speed and win rate, highlighting the advantage of explicitly modeling higher-order interactions among agents.
>
>     ### Table 2. Performance comparison with NA$^2$Q on corridor and 8m_vs_9m.
>
>     | Method        |     8m_vs_9m    |     corridor  |
>     | :------------:| :---------------: | :---------------: |
>     | VDN           |   0.80$\pm$0.11   |   0.46$\pm$0.36   |
>     | QMIX          |   0.75$\pm$0.11   |   0.01$\pm$0.01   |
>     | NA$^2$Q       |   0.83$\pm$0.07   |   0.63$\pm$0.20   |
>     | QCoFr (Ours)  |   **0.93$\pm$0.02**   |   **0.79$\pm$0.03**   |
>
>
> 5. **Stability with depths.**
>
>    **Response:** Please refer to the response to **Reviewer CAhf**.
>
> 6. **ReLU in the credit assignment mechanism.**
>
>     **Response:** We appreciate your insightful question regarding interpretability in the context of credit assignment and activation choices. Credit assignment mechanism relies on monotonic linear layers with non-negative weights, which aggregate outputs from different levels of interaction modeled by the continued fraction network. Each layer in CFN captures a distinct interaction order—1-order (individual), 2-order (pairwise), and so on. Because CFN approximates a power series, the credit assignment weights can be interpreted analogously to Taylor coefficients, which indicate how much each coalition contributes to the team. Besides, the use of ReLU is not central to interpretability per se but serves as a computationally efficient monotonic nonlinearity. The work [R1] shows that replacing softmax with normalized ReLU in attention mechanisms reduces computation and improves stability. Furthermore, we acknowledge that other monotonic activation functions (e.g., Softplus, Sigmoid) could also be used in the credit assignment module without compromising interpretability. Softplus, in particular, offers a smooth approximation of ReLU and may facilitate optimization in some settings. We chose ReLU for its simplicity and empirical performance, but the framework is compatible with these alternatives.
>
>    [R1] Mitchell Wortsman, et al., Replacing softmax with relu in vision transformers. arXiv preprint arXiv:2309.08586, 2023.
>
> 7. **Move toward fully decentralized approaches.**
>
>    **Response:** Thank you for the insightful comments. QCoFr follows the CTDE paradigm that agents have access to global information during training but rely on local policies during execution. It may be a challenge to simply move to fully decentralized learning, where each agent learns without a central critic. Without the centralized training, agents cannot directly optimize joint objectives. One possible solution is to employ decentralized policy gradient methods with local critics that exchange messages subject to a bandwidth constraint. The VIB could be extended to learn compact messages. Another possibility is to use implicit coordination via mutual information regularization, encouraging agents' policies to align with latent team variables.
>
> 8. **Variable definitions and typos.**
>
>    **Response:** We appreciate the feedback regarding undefined variables.  In the next revision, we will: 1) Clearly define all symbols when first used. 2) Move Definition 1 and related mathematical preliminaries to the appendix to free space in the main text for clearer definitions of the main quantities. 3) We will also correct the grammatical issues identified around lines 30, 45, 73, and 187.

---

> > ### Comment · Reviewer_fozS · 2025-08-05
> >
> > So regarding the depth issue and instability, in your experiments with 6h_vs_8z, what was the depth? and what happens when you increase it to one more level? Also, what do you think is missing in your methodology so that the score in such a task is low?

---

> ### Author Response · Authors · 2025-08-05
> **Regarding the depth issue and instability**
>
> Thank you for this insightful follow-up discussion. We believe this question touches on a core design of our method, and we appreciate the opportunity to clarify.
>
> For the 6h_vs_8z scenario, we set the interaction depth $d$ to 6, equal to the number of allied agents (see Table 3 in the appendix for detailed settings). The depth $d$ of CFN denotes the highest order of agent interactions modeled. In QCoFr, this directly determines the largest coalition size considered when decomposing the joint value function. In theory, increasing $d$ allows representation of more complex multi-agent cooperation patterns, potentially improving coordination and credit assignment by capturing joint behaviors that lower-order models miss. However, if $d$ exceeds the true interaction complexity of the task (i.e., how many agents actually need to coordinate), the model may start fitting spurious higher-order correlations. This can slow down learning or even degrade performance due to overfitting to non-existent interaction patterns. Empirically, our SMAC experiments support this: as shown in Figure 6, performance improves as $d$ increases up to the team size, but using $d$ beyond the team size (e.g., $d=8$ for a 6-agent team) leads to a slight performance drop and slower convergence. We attribute this degradation to the model trying to fit 7- or 8-agent “interactions” that do not actually exist in a 6-agent team, introducing redundant parameters that hinder learning. In short, for a 6-agent task, $d=6$ is optimal, while $d=8$ added computational overhead and overfitting risk with no benefit. Similarly, in the 3s_vs_5z scenario (team of 3), using $d=4$ slowed early learning, confirming that depth beyond the team size can introduce noise and slow convergence.
>
> Regarding the lower score on the 6h_vs_8z scenario, this scenario is indeed extremely challenging due to its asymmetry (6 fragile hydralisks versus 8 durable zealots). Hydralisks have low health and are highly vulnerable in close combat, whereas Zealots are tougher melee units that are hard to eliminate with individual attacks. To succeed, the agents must learn complex coordination strategies, such as forming tactical positions based on armor types, inducing enemy units to chase while maintaining optimal distance to minimize damage (i.e., kiting), focusing fire to eliminate enemy units one by one, and synchronizing their movements to launch attacks from multiple directions or utilize terrain advantages. These behaviors require agents to reason about the global context, anticipate teammates’ actions, and adapt to enemy strategies—all under partial observability and without inter-agent communication. We believe that a more efficient exploration mechanism or explicitly promoting policy diversity would help the agents discover these complex behaviors, potentially improving performance on such a difficult scenario.

---

> > ### Comment · Reviewer_fozS · 2025-08-06
> >
> > Thank you for the clarification. Please include a summary of these details in your latest version in the conclusion or limitations section.

---

> > > ### Author Response · Authors · 2025-08-06
> > >
> > > We appreciate your confirmation that our clarifications address your concerns. As suggested, we will add a concise summary of these details—including discussion of the depth–stability of QCoFr trade-offs and a clearer differentiation from NA$^2$Q—to the conclusion/limitations section in the revised manuscript. Thank you again for the constructive feedback and for helping us improve the paper.

---

### Official Review · Reviewer_CAhf · 2025-07-01

**Clarity:** 2
**Significance:** 2
**Originality:** 2
**Rating:** 4
**Confidence:** 4

**Summary:**

This paper proposes a new method named QCoFr that uses higher order interactions inspired by Taylor series to improve the decomposition of joint Q functions in a manner that is more interpretable.

**Questions:**

Please find some questions below.
1. the architecture of the proposed method seems quite close to NA2Q's architecture; could the authors elaborate better how their method is different? this could be made more clear in the paper
2. what is the intuition behind predicting the joint optimal action from the history of observations (as shown in figure 2)? what information can that give to learn a good value for $m$?
3. there are several claims regarding better credit assignment across the paper; however, from my understanding there is not clear evidence that the credit is being correctly assigned as a product of the proposed method; i can find the contribution levels illustrated, but are these reflected in the credit given to the agents?
4. since the proposed method is closely aligned with one of the related methods Na^2Q, it would be interesting to see how they compare in terms of interpretability, since the authors also mention that this method also enables interpretability (lines 89-92); has this been tried?
5. in figure 9, the q-values similarities between QMIX and QCoFr are quite close; while when comparing to VDN the difference is clear, with QMIX we could argue that QCoFr is not necessarily producing that much diverse behaviours when compared to QMIX; could the authors elaborate on this?
6. in section 5.2, in the interaction orders ablation: "These findings demonstrate that modelling higher-order interactions yields performance improvements for complex coordination tasks, but only within a bounded regime that aligns with the task’s intrinsic interaction complexity" (lines 270-273) - this is an interesting observation but it shows that for different environments the best depth level can be different; have the authors further explored these findings? is it possible to theoretically define this bounding that can create a balance between performance improvements and scalability?

**Ethical Concerns:**

["NO or VERY MINOR ethics concerns only"]

**Final Justification:**

Most of unclarities were addressed by the authors, hence I have reflected that in my score.

**Limitations:**

These have been discussed in the conclusion and are appropriate. Please also refer to my comments above for other potential limitations.

**Quality:**

3

**Strengths And Weaknesses:**

Overall, this paper is well written and generally well structured. The proposed method shows very strong performances in different tasks, outperforming some of the current SOTA methods in some of the environments tested. The method is also theoretically grounded.

However, there are some less clear points that I list below; specifically, I am also a bit confused regarding the novelty of this work when compared to the previous NA2Q [1] that is mentioned in this paper.

1. There are several losses represented in figure 2 but it is not clear how some of these losses are combined and their contribution/exact meaning (CE loss for example)
2. The proofs in the appendix are not presented in a very intuitive manner; for example, after reading section 3 and definition 1, the authors refer to a proof but when we check the appendix this proof is difficult to quickly find; I suggest the authors could clarify where in the appendix the proofs are in general
3. in the end of section 3, two variations of the proposed method are mentioned: CFN-D and CFN-C; however, in the main paper it is not clear how these are different from normal CFN
4. the architecture of the proposed method seems close to the NA2Q's architecture and it is not very clear how both methods clearly differentiate from each other

### Minor:
* in figure 9, it would be good to have the titles of the subfigures indicating to which baseline the matrices correspond to

[1] https://arxiv.org/abs/2304.13383

---

> ### Author Rebuttal · Authors · 2025-07-31
>
> We sincerely thank you for your time in providing detailed feedback on the paper to better highlight the key contributions and novelties of the paper. We detail our responses to the questions below.
>
> 1. **Relation to NA$^2$Q and novel contributions.**
>
>    **Response:** Thank you for raising this important point. While our method shares the high-level goal of modeling interactions among agents with NA$^2$Q, the architectural design and core contributions differ significantly:
>
>    NA$^2$Q uses the generalized additive models (GAM) to enumerate all possible agent coalition terms up to the full interaction order, incurring exponential complexity $\sum_{i=1}^n C_n^i = 2^n - 1$. This becomes infeasible in large-scale or high-order coordination settings. In contrast, QCoFr introduces a CFN whose depth $d$ corresponds directly to the interaction order, enabling explicit higher-order interaction modeling with linear computational complexity $O(n)$. To our knowledge, QCoFr is able to provide high-order cooperation patterns and tractable learning of arbitrary-order interactions in linear time. Regarding the advantage of QCoFr, please see the details of the response to **Reviewer PxnY**.
>
>    QCoFr adopts a VIB to encode assistive information $\boldsymbol{m} \sim q(\boldsymbol{m}|\boldsymbol{h})$ predicting $\boldsymbol{u}^*$, whereas NA$^2$Q relies on a VAE-based module for capturing latent semantics. This improves credit estimation by selectively filtering out confounding factors in partial observability, aligning the encoding with optimal decision-making. We further evaluated NA$^2$Q, with its VAE module replaced by our VIB design and observed consistent performance gains, reinforcing the benefit of our approach.
>
> 2. **Intuition behind predicting the joint optimal action.**
>
>    **Response:** The goal is to address the credit assignment problem in cooperative MARL: how to infer the contribution of each agent or coalition to the team under partial observability. When agents condition their utilities on the global state $\boldsymbol{s}$, there is a backdoor path  $\boldsymbol{s} \leftarrow E \rightarrow \tau \rightarrow Q_{tot}$, where $E$ denotes unobserved environment dynamics and $\tau$ an agent’s history. This path can induce spurious correlations, causing the mixing network to factorize credit based on confounders rather than genuine causal effects. To break this path, we follow the principle used in other works, such as PAC [R1] and IMAC [R2], generate assistive information by predicting the joint optimal action $\boldsymbol{u^*}$ from each agent's history and then encode it via VIB.
>
>    In detail, the assistive encoder takes the local history $h_i$ and outputs a latent variable $m_i$. We train this encoder with a cross‑entropy loss to predict the joint optimal action $\boldsymbol{u^\*}$; at the same time, a KL‑divergence term constrains the mutual information $I\left(\boldsymbol{m}, \boldsymbol{u}^\* ; \boldsymbol{\phi}\right)$.  This forms a Markov chain $\boldsymbol{h}-\boldsymbol{m}-\boldsymbol{u^\*}$.  The intuition is that if $\boldsymbol{m}$ can predict $\boldsymbol{u^\*}$ well, it must encode precisely those features of the agent's history that are relevant for selecting the optimal joint action, while the information bottleneck discourages encoding irrelevant details. This helps the mixing network facilitate more accurate credit assignment conditioning on $\boldsymbol{m}$ along with the global state $\boldsymbol{s}$ whose local histories are genuinely informative for the joint decision, and those whose histories are noisy or misleading. We will highlight this causal perspective and the connection to the PAC framework in the revised manuscript.
>
>    [R1] Hanhan Zhou, et. al., PAC: Assisted value factorisation with counterfactual predictions in multi-agent reinforcement learning, NeurIPS, 2022.
>
>    [R2] Rundong Wang, et. al., Learning efficient multi-agent communication: an information bottleneck approach, ICML, 2020.
>
> 3. **Credit Assignment.**
>
>    **Response:** In our work, the notion of *contribution* refers to the product of each agent’s individual Q-value (or multi-agent interaction term) and the credit assigned to it. For example, a third-order interaction among agents $i, j, k$ is represented as $Q_i Q_j Q_k$, and its contribution is defined as $\beta_{ijk} Q_i Q_j Q_k$, where $\beta_{ijk}$ denotes the assigned credit. Thus, there is a direct and interpretable correlation between the contribution level and the credit allocation for each agent and their interactions. Our method explicitly models $Q_{tot}$ as a sum of contributions from individual agents and their interaction terms through a continued fraction structure, which—together with the global state $\boldsymbol{s}$ and assistive information—enables the learning of fine-grained credit assignment. As shown in Figure 8, increasing the interaction order in the model leads to better representation of cooperative behavior among agents, and we observe that higher-order interaction terms exhibit significantly increased contribution levels. This behavior aligns with the agents' observed coordination, demonstrating that our method provides a more accurate and interpretable credit assignment than those considering only low-order interactions.
>
>
> 4. **Choosing the depth (interaction order).**
>
>    **Response:** Thank you for raising this insightful point. Our ablation in Section 5.2 shows that the performance degrades as the interaction order increase beyond a certain depth. One key observation is that when the order of interactions exceeds the number of agents, e.g., considering interactions beyond the $n$-th order for $n$ agents, it leads to impractical terms ($n+1$-order or higher-order interactions) that cannot be realized by the agents. These unnecessary higher-order terms may still be assigned weights during training, but they are meaningless and may overfit or propagate noise, potentially affecting the convergence speed. Generally, each additional fraction layer captures one higher order of agent interaction. The optimal depth should depend on the intrinsic cooperation complexity of the environment (e.g., pairwise coordination vs. five agent synchronization).  We have not yet derived a theoretical bound on the optimal depth, but we are investigating adaptive techniques to learn the appropriate depth automatically, akin to halting criteria in adaptive computation. We will discuss these ideas in the revised conclusion.
>
> 5. **Explanation of Figure 9.**
>
>    **Response:** We appreciate your careful observation. Although the Q-values similarities between QMIX and QCoFr appear close in Figure 9, QCoFr actually produces more diverse behaviors compared to QMIX. From it, we can observe that in QMIX, the cosine similarity between the Q-values of other agents and the current agent’s Q-values ranges from 0.8 to 1, while in our method (QCoFr), the range is from 0.61 to 1. This indicates a significantly larger variance in the Q-values between agents in QCoFr, leading to more diverse behaviors. Furthermore, as shown in Figure 8, QCoFr exhibits more complex collaborative behaviors, which also reflect the increased diversity in its actions. At a given moment, agents may collaborate and take similar actions, which could not always be the case if the Q-values of all agents differ significantly. However, overall, QCoFr demonstrates more diverse behaviors compared to QMIX. We will clarify this interpretation in the revised manuscript.
>
> 6. **Loss terms in Figure 2.**
>
>    **Response:** Thank you for your careful comment, and we apologize for the oversight in not providing a clear explanation of the losses. The CE Loss and KL Loss shown in Figure 2 are both part of the VIB Loss (Equation 11). Specifically, the Cross-Entropy (CE) Loss ($\mathcal{L}\_{CE}$) is defined as $\mathcal{L}\_{CE} = \frac{1}{N} \sum_{i=1}^N \mathbb{E}\_{\epsilon \sim p(\epsilon)} \left[-\log q(u_i^* \mid f(h_i, \epsilon)) \right]$, and the Kullback Leibler (KL) Loss ($\mathcal{L}\_{KL}$) is defined as $\mathcal{L}\_{KL} = \mathrm{KL}\left[p(\boldsymbol{m} \mid h_i), \tilde{q}(\boldsymbol{m})\right]$. Thus, the overall VIB Loss can be expressed as $\mathcal{L}\_{VIB} = \mathcal{L}\_{CE} + \beta \cdot \mathcal{L}\_{KL}$. In the context of the VIB framework, the two losses serve different purposes:
>
>    A. The CE Loss encourages the model to make accurate predictions by minimizing the difference between the predicted and target distributions, ensuring the learned representations are relevant for task performance.
>
>    B. The KL Loss serves as a regularizer, which aims to encourage the model to make the posterior distribution of the latent variable $\boldsymbol{m}$ close to the prior distribution $\tilde{q}(\boldsymbol{m})$. This controls the flow of information, preventing the network from retaining redundant or irrelevant information, thereby enforcing the information bottleneck.
>    We will add explicit equations and describe the combination strategy for clarity.
>
> 7. **Proofs in the appendix are not presented in a very intuitive manner.**
>
>    **Response:** We apologise for the difficulty in locating proofs.  We will revise the paper to insert cross references after each definition and theorem so that readers can quickly find the corresponding proof.
>
> 8. **Minor Points.**
>
>    **Response:**
>
>    1) We agree that adding subfigure titles to Figure 9 will improve clarity and will revise accordingly.
>
>    2) We will refine the manuscript around CFN-D and CFN-C to make their differences explicit.

---

> > ### Comment · Reviewer_CAhf · 2025-08-04
> > **Thanks**
> >
> > Thanks I have no further questions, on the assumption that the clarifications will be included in the revised version.

---

> > > ### Author Response · Authors · 2025-08-05
> > >
> > > We sincerely appreciate your thoughtful feedback and are pleased that our response has successfully addressed your concerns. We'll make sure to incorporate the relevant clarifications and discussions into the revised version of the paper.

---

### Official Review · Reviewer_PxnY · 2025-07-03

**Clarity:** 3
**Significance:** 2
**Originality:** 3
**Rating:** 4
**Confidence:** 4

**Summary:**

This paper proposes a novel MARL value-decomposition paradigm. By introducing continued fraction modules, the authors achieve two key improvements: reducing computational complexity for higher-order terms and realizing interpretable value decomposition. Furthermore, the authors construct a practical value decomposition approach named QCoFr by integrating Continued Fraction Network (CFN) and Variational Information Bottleneck (VIB) modules. Experimental results demonstrate the superiority of QCoFr compared to existing value-based methods, supported by ablation studies and interpretability experiments.

**Questions:**

In addition to the concern mentioned above, I would like the authors to address the following questions:

1.Linking value decomposition with agent interactions feels somewhat counterintuitive. Do the authors have any deeper insights on this?

2.Why was Qatten not mentioned in the method discussion, given that it also implements a Taylor-like decomposition among agents? This is also an interpretable algorithm. I suggest the authors analyze the complexity of the terms in this algorithm and conduct a thorough comparison (beyond experiments) with it.

**Ethical Concerns:**

["NO or VERY MINOR ethics concerns only"]

**Final Justification:**

Most of my concerns addressed.

**Limitations:**

Yes.

**Paper Formatting Concerns:**

No related concerns at this time.

**Quality:**

3

**Strengths And Weaknesses:**

**Strengths**:

(Major) From an innovation perspective, the authors propose a novel, interpretable value decomposition method and provide a universal approximation theorem proof under the MARL framework.

(Major) In terms of writing, the paper is well-written with a clear structure. The illustrations are precise and facilitate comprehension and reading.

(Minor) The comparative experiments are comprehensive, with thorough baseline and benchmark selections.

---

**Weaknesses**: I will consider recommending acceptance if the authors satisfactorily address my concerns:

(Major) I am particularly perplexed by the role of the VIB model in the overall methodology and paper. From the perspective of proposing a novel value decomposition paradigm, the CFN model appears sufficient. The VIB model seems unnecessary in this paper, contributing neither to interpretability nor computational complexity reduction, and potentially disrupting the reading experience. Moreover, the ablation studies suggest that the VIB model does not decisively improve performance. Consequently, I reasonably suspect the VIB model might be artificially introduced to inflate innovation claims.

(Major) Why does the Mixing method still adhere to the IGM principle (is this a regression)? Numerous studies have already explored value decomposition without IGM constraints, demonstrating superior performance on complex tasks compared to QMIX and VDN. Theoretically, removing IGM constraints should enhance representational capacity. I recommend the authors address either: a. the necessity of maintaining IGM (e.g., theoretical integration with CFN) or b. the applicability of the proposed method under non-IGM conditions.

(Major) While the initial sections of the paper sparked my interest in the method's interpretability, the experimental results fail to adequately demonstrate its superiority in this dimension. What is the authors' definition of interpretability? The experiments could be more comprehensive, such as discussing order-related influences or providing more extensive visualizations. Overall, the interpretability experiments are insufficient. (Note: I personally believe that the visualization analysis of diversity is not strongly related to the interpretability of the proposed method. This analysis only demonstrates that the proposed method can learn diverse strategies more quickly in certain scenarios, but the method itself is not inherently related to diversity.)

---

> ### Author Rebuttal · Authors · 2025-07-31
>
> Dear Reviewer,
>
> We appreciate your insightful comments and provide our response as follows.
>
> 1. **The role of the VIB.**
>
>    **Response:**
>    We would like to emphasize that VIB is an important module in our method by promoting credit assignment in cooperative MARL. Credit assignment aims to attribute global team success to individual agent contributions. Due to partial observation, there exists a potential backdoor path $\boldsymbol{s} \leftarrow E \rightarrow \tau \rightarrow Q_{tot}$ between the global state $\boldsymbol{s}$ and $Q_{tot}$, which can lead to a spurious correlation that causes confounding bias for estimating credit assignment. Following prior works [R1, R2], we address this by intervening on the global state $\boldsymbol{s}$ using local history $\tau_i$ via VIB. To this end, we encode the agent's history $\boldsymbol{h}$ into a latent representation $\boldsymbol{m}$, forming a Markov chain $\boldsymbol{h} \rightarrow \boldsymbol{m} \rightarrow \boldsymbol{u}^*$, where $\boldsymbol{m}$ is optimized to be maximally informative for predicting the optimal joint action. This can encourage $\boldsymbol{m}$ to memorize $\boldsymbol{u}^\*$ with meaningful information. This latent $\boldsymbol{m}$ acts as assistive information during value factorization to reduce confounding and promote policy learning.
>
>    Empirically, while the gains from VIB appear modest on simple tasks, it accelerates the early stage and becomes critical in harder scenarios. On the super-hard map 6h_vs_8z, removing VIB results in a significant performance drop, indicating the necessity of additional latent information for complex coordination. As shown in Table 1, across one hard and two super-hard tasks, QCoFr with VIB achieves better results, which confirms the effectiveness of leveraging additional information.
>
>     ### Table 1. Additional comparison on 3 extra scenarios.
>
>     | Method        |     8m_vs_9m      |       corridor    |  3s5z_vs_3s6z     |
>     | :------------:| :---------------: | :---------------: | :---------------: |
>     | VDN           |   0.80$\pm$0.11   |   0.46$\pm$0.36   |   0.64$\pm$0.33   |
>     | QMIX          |   0.75$\pm$0.11   |   0.01$\pm$0.01   |   0.61$\pm$0.19   |
>     | QCoFr w/o VIB |   0.82$\pm$0.07   |   0.60$\pm$0.32   |   0.65$\pm$0.25   |
>     | QCoFr (Ours)  |   **0.93$\pm$0.02**   |   **0.79$\pm$0.03**   |   **0.87$\pm$0.07**   |
>
>
>
>    [R1] Jiahui Li, et. al., Deconfounded value decomposition for multi-agent reinforcement learning, ICML, 2022.
>
>    [R2] Zichuan Liu, et. al., NA$^2$Q : Neural attention additive model for interpretable multi-agent Q-learning. ICML, 2023.
>
> 2. **IGM Constraint.**
>
>    **Response:** Our goal in this work is to introduce a novel high‑order value decomposition paradigm via continued fractions. Retaining the IGM constraint allows us to isolate the benefits of the continued fraction structure without conflating them with non‑monotonic search mechanisms. The universal approximation theorem we prove applies to any linear combination of continued fractions; it does not rely on non‑negative weights, so the theory could extend to non‑IGM mixers. Integrating CFN with fully IGM‑free methods like DAVE is an interesting direction [R3], but it is outside the scope of this submission. DAVE points out that value decomposition methods typically operate under IGM, which links the optimal joint action to the optimal individual action. Removing IGM requires agents to search for the global best action at execution time and often entails an additional network.
>
>    To probe this, we conducted additional experiments relaxing the IGM constraint. Results in Table 2 show QCoFr achieves comparable or slightly improved performance without IGM, indicating that our architecture can still recover high-quality joint actions. We will make this discussion explicit in the revised paper.
>
>     ### Table 2. Comparison with and without IGM constraint.
>
>     | Method        |     2c_vs_64zg    |     3s5z_vs_3s6z  |      6h_vs_8z     |
>     | :------------:| :---------------: | :---------------: | :---------------: |
>     | VDN           |   0.88$\pm$0.05   |   0.64$\pm$0.33   |   0.00$\pm$0.00   |
>     | QMIX          |   0.96$\pm$0.05   |   0.61$\pm$0.19   |   0.11$\pm$0.19   |
>     | QCoFr w/o IGM |   0.96$\pm$0.04   |   **0.89$\pm$0.07**   |   **0.67$\pm$0.07**   |
>     | QCoFr (Ours)  |   **0.96$\pm$0.03**   |   0.87$\pm$0.07   |   0.58$\pm$0.12   |
>
>
>    [R3] Zhiwei Xu, et. al., Dual self-awareness value decomposition framework without individual global max for cooperative MARL. NeurIPS, 2023.
>
> 3. **Additional interpretability experiments.**
>
>    **Response:** Thank you for prompting further clarification on interpretability. Our motivation is to design an interpretable model where agent interactions can be easily understood. The continued fraction structure of CFN enables symbolic representation of the Q-function, which can be analytically expanded into a multivariate power series using tools such as *Mathematica*. This expansion yields exact attributions for both individual agent terms and higher-order interactions.
>    On the 5m_vs_6m scenario (in Figure 8), a CFN with a shallow depth $d=2$ produces the symbolic expression:
>
>    $$ \alpha_1 \cdot \frac{1}{0.2455 Q_1 + 0.3093 Q_2 + 0.0015 Q_3 + 0.2326 Q_4 + 0.0459 Q_5 - 1.1213 + \frac{1}{0.0977 Q_1 +         0.2005 Q_2 + 0.2859 Q_3 + 0.2732 Q_4 + 0.1135 Q_5 + 0.4600}} + \alpha_2 \cdot \frac{1}{0.2277 Q_1 + 0.3581 Q_2 + 0.0329 Q_3 + 0.1803 Q_4 + 0.0367 Q_5 + 1.4818 - \frac{1}{0.2311}} +   f_0,$$
>
>    where $\alpha_k$ denotes the attention weight assigned to each ladder varing over time. Expanding this at $t=10$ into a power series up to 2-order yields:
>
>    $$Q_{tot}= 0.08Q_1+0.26Q_2+0.60Q_3+0.46Q_4+0.22Q_5-0.06Q_1Q_2-0.01Q_1 Q_3-0.05Q_1Q_4-0.01Q_1Q_5+0.21Q_2Q_3+0.07Q_2Q_4+0.06Q_2Q_5+0.50Q_3Q_4+0.27Q_3Q_5+0.16Q_4Q_5+\cdots.$$
>
>    We observe that the terms with the largest coefficients, such as $Q_3Q_5$,$Q_3Q_4$,$Q_2Q_3$, align well with the most critical agent combinations shown in Figure 8.
>    For a deeper CFN with $d=4$, we omit the full expression due to space constraints
>
>     $$\frac{1}{0.2804 Q_1 +  \cdots + 0.4022 Q_5 - 2.0528+\frac{1}{0.0896 Q_1 +  \cdots + 0.0590 Q_5 - 0.1697 + \frac{1}{0.3713 Q_1 + \cdots + 0.2525 Q_5 - 0.2219+\frac{1}{0.2575 Q_1 + \cdots + 0.3108 Q_5 + 0.3811}}}}\cdots .$$
>
>    At $t=10$, we extract the top-10 terms (by coefficient magnitude):
>
>     | term     | coefficients | term | coefficients |
>     | --------- | --------------- | ----- | ------ |
>     |  $Q_2 Q_3$ | 0.579 | $Q_1 Q_2 Q_3 Q_5$ | 0.449 |
>     |  $Q_2 Q_3 Q_4$ | 0.579 | $Q_3 Q_4 Q_5$ | 0.447|
>     |  $Q_1 Q_2 Q_4$ | 0.567 | $Q_1  Q_3 Q_5$ | 0.375 |
>     |  $Q_2 Q_4 Q_5$ | 0.543 | $Q_1 Q_4 Q_5$ | 0.123 |
>     |  $Q_1 Q_3 Q_4$ | 0.469| $Q_1 Q_2 Q_5$ | 0.098 |
>
>    This result explicitly quantifies the contribution of higher-order interaction terms to their team, most of which align with the top-ranked agent combinations highlighted in Figure 8. In contrast, prior methods limited to unary or pairwise terms fail to capture these complex yet key patterns of cooperation with more than 3-order interactions.
>
> 4. **The connection between value decomposition and agent interactions.**
>
>    **Response:** Thank you for highlighting this point. We believe that treating value decomposition as a structure of interaction orders is both theoretically justified and practically advantageous. In cooperative MARL, the joint Q-function encodes not only the individual contribution of each agent but also the surplus value generated when agents cooperate. It resembles Shapley value and synergy decomposition: first‑order terms capture individual effects, while higher‑order terms quantify the effects of coalitions [R4]. Such an interaction modeling helps avoid over‑ or under‑estimating any agent's contribution and provides a clear credit assignment estimation—interaction terms reflect cooperation patterns. VDN considers 1-order interaction and takes a sum of individual values, which may limit its capacity to ignore higher-order interactions. QMIX theoretically can represent any order interaction relationship among agents with two non-linear networks. QCoFr theoretically not only can own the same capacity as QMIX but also explicitly model all possible higher-order interactions with linear‑time complexity.
>
>    [R4] Jianhong Wang, et al., SHAQ: Incorporating Shapley Value theory into multi-agent Q-learning. NeurIPS, 2022.
>
>
> 5. **Discussion of Qatten.**
>
>    **Response:** We agree that Qatten is a related baseline. We will add it in the revision and summarize the key differences here:
>
>    • Complexity:  Qatten relies on multi‑head attention, where each head computes a full similarity matrix over all agents, yielding $O(n^2)$ computational cost. In contrast, our CFN recursively aggregates individual utilities, leading to $O(n d)$ complexity where $d$ is the depth.
>
>    • Interpretability:  In Qatten, attention weights indicate agent importance, but there is no direct correspondence between heads and interaction orders. QCoFr explicit models of higher‑order interactions avoid the combinatorial explosion.
>
>    • Performance: We have conducted extended experiments on SMAC, where QCoFr achieves higher gains than Qatten on a range of tasks, which indicates the superiority of considering high-order interactions.
>
>     ### Table 3. Performance Comparison with Qatten on 3  maps.
>     | Method        |     5m_vs_6m      |    3s5z_vs_3s6z    |    6h_vs_8z   |
>     | :------------:| :---------------: | :---------------: | :---------------: |
>     | VDN           |   0.64 ± 0.06   |   0.64 ± 0.33   |   0.00 ± 0.00   |
>     | QMIX          |   0.66 ± 0.16   |   0.61 ± 0.19   |   0.11 ± 0.19   |
>     | Qatten        |   0.64 ± 0.13   |   0.04 ± 0.04   |   0.01 ± 0.03   |
>     | QCoFr (Ours)  |   **0.78 ± 0.09**   |   **0.87 ± 0.07**   |   **0.58 ± 0.12**  |

---

> > ### Comment · Reviewer_PxnY · 2025-08-06
> >
> > Thank you for the authors' response.
> >
> > I am satisfied with your reply and you have addressed almost all of my concerns.
> >
> > I need to further confirm the rationality of the additional experiments and the interpretability.
> >
> >  1. Regarding the part about relaxing IGM, how exactly was the experiment conducted? What are the specific algorithm and implementation details?
> >
> > 2. Under the settings of relaxed IGM, how will the algorithm's interpretability change?
> >
> > I have already adjusted my score and look forward to the authors' further response.

---

> > > ### Author Response · Authors · 2025-08-06
> > >
> > > We sincerely appreciate your positive acknowledgment of our responses to your valuable comments, as well as your adjustment of the rating. Thank you for providing us the opportunity to further clarify the remaining questions regarding the implementation details of relaxing the IGM constraint.
> > >
> > > **Regarding the implementation details of relaxing the IGM constraint.** To realize the implementation of relaxing the IGM constraint in our CFN-based mixing network, we modified the activation function in each CFN layer by removing the non-negativity restriction (allowing negative contributions). In the original IGM-compliant design, each CFN layer uses an activation $f(z) = \frac{1}{\max(|z|,\delta)}$ (with $\delta$ a small positive constant) to ensure outputs are non-negative and thus the mixing is monotonic in the individual $Q$-value. We replaced this with $f(z) = \operatorname{sgn}(z) \cdot \frac{1}{\max(|z|,\delta)}$. Here, $\operatorname{sgn}(z)$ is the sign of the pre-activation $z$ (which is the sum of the current layer’s weighted input and the reciprocal of the previous layer’s output), so $f_{\text{relaxed}}(z)$ retains the same magnitude $1/\max(|z|,\delta)$ but allows positive or negative signs. This minimal change lifts the strict monotonicity requirement while leaving all other aspects of the CFN architecture and training procedure unchanged. We emphasize that no special hyperparameter tuning or scenario-specific adjustments were needed for this “relaxed IGM” version – we can simply apply it to the same benchmark tasks. In practice, this change expanded the expressiveness of the mixing network, enabling it to represent a broader class of value decomposition functions (including those with diminishing or negative interaction effects). And, we indeed observed a consistent performance improvement in our experiments when using the relaxed activation during the rebuttal stage. We will clarify these implementation details in the revision to validate the rationale of this additional experiment.
> > >
> > > **Interpretability under the relaxed IGM setting.** Importantly, the setting of relaxed IGM does not compromise the interpretability of our method. The CFN mixing network still expresses the joint action-value as an explicit linear combination of individual and multi-agent interaction terms (continued-fraction expansion) – the only difference is that some terms’ coefficients can now be negative. The output of the relaxed CFN can be expanded (via power-series expansion) in the same way as before: it remains a sum of products of individual $Q_i$ values up to $d$-th order. In other words, we still obtain a symbolic formula for $Q_{\text{tot}}$ showing each agent’s contribution and each coalition’s contribution; the structure is identical to the IGM-constrained case, except that those contribution coefficients $c_{p_1…p_n}$ are no longer restricted to be non-negative. This means the intrinsic interpretability is preserved – we can read off the learned coefficients to understand how each agent or coalition of agents contributes to their team. The relaxed CFN output is thus a transparent formula consisting of interpretable components (individual agent terms and high-order coalition terms), just as before. In summary, relaxing the IGM constraint expands the space of learnable functions but retains the clear continued-fraction structure that allows us to attribute credit to individual agents and agent coalitions. We will include a symbolic example like the one above in the final version to illustrate that the interpretability of our method remains intact under the relaxed setting.

---

### Comment · Area_Chair_4wyL · 2025-08-05

Dear Reviewers

Thanks for participating in discussion. The author-reviewer discussion period is extended to Aug. 8, 11:59PM AoE.

If you have not yet, I kindly ask you to

- Read the author rebuttal and other reviews.
- Post your response to the author rebuttal as soon as possible.
- Engage in discussion with the authors and other reviewers where necessary.

It is not permitted by the conference policy that a reviewer submits mandatory acknowledgement without a single reply sentence to author rebuttal.

If you have already engaged in author-reviewer discussion, I sincerely thank you. If not, I kindly ask you to engage in discussion.

Thank you very much for your help.

Best regard,

AC

---

### Comment · Area_Chair_4wyL · 2025-08-08

Dear Reviewers

Thank you again for participating in reveiwer-author discussion.

The reviewer-author discussion is coming to an end soon. If you did not submit mandatory acknowledgement yet, please submit the ack, final score and justification by the end of the reviewer-author discussion period.

Thanks.

Area Chair

---

### Note · Authors · 2025-08-14

We appreciate all reviewers for their positive assessment and recognition of our contributions. We are pleased that their concerns have been resolved and no further questions have been raised. Below we summarize the clarifications and additions that will appear in the revised manuscript.

**Role of VIB.** We clarified that VIB is used to break a backdoor path under partial observability and to distill history features predictive of the joint-optimal action $\boldsymbol{u^*}$. This improves credit estimation in the mixer. New additional ablations have shown consistent gains on super-hard maps (e.g., corridor, 3s5z_vs_3s6z).

**Interaction orders and depth.** Section 5.2 shows that performance improves as interaction depth increases up to the task’s intrinsic coordination level, but degrades (or learns more slowly) when depth exceeds the number of agents. When modeling ($n+1$-order (or higher) that cannot be realized by the agents. These unnecessary higher-order terms may still be assigned weights during training, but they are meaningless and may overfit or propagate noise. The optimal depth should depend on the intrinsic cooperation complexity of the environment. Following Reviewer WW9h's suggestion, we also report results on MACO, where QCoFr achieves better performance than VDN, QMIX, and NA$^2$Q, supporting our claims about scaling with interaction orders.

**Relation to NA$^2$Q and novel contributions.** We add a clear contrast: NA$^2$Q models coalition terms (exponential, $\sum_{i=1}^n C_n^i = 2^n - 1$), whereas our CFN achieves linear complexity $O(n)$ with a depth–order correspondence. QCoFr is the first method to achieve symbolic interpretability and tractable learning of arbitrary-order interactions in linear time. Moreover, QCoFr employs VIB to encode assistive information  $\boldsymbol{m} \sim q(m|\boldsymbol{h})$ predicting $\boldsymbol{u^\*}$, whereas NA$^2$Q relies on a VAE-based module for capturing latent semantics. This improves credit estimation by selectively filtering out confounding factors in partial observability, aligning the encoding with optimal decision-making. We further evaluated NA$^2$Q, with its VAE module replaced by our VIB design and observed consistent performance gains, confirming the advantage of our method.

We will incorporate these clarifications so that the final version is clearer, better grounded, and easier to assess. Thank you again for the thoughtful feedback that strengthened the paper.

---

### Decision · Program_Chairs · 2025-09-17

**Decision:**

Accept (poster)

**Comment:**

This paper proposes a new value decomposition architecture for MARL capturing higher order interaction among agents based on continued fractional neural networks. The new method achieves noticeable performance gain in difficult SMAC tasks such as 3s5z_vs_3s6z, showing such hard tasks indeed require higher order interaction modeling. In the research line of value decomposition under CTDE for multiagent reinforcement learning, it seems that the proposed method makes a relevant position.

This work is basically based on the previous work of Puri et al.'s CoFrNets.  Clearly, cite this in Section 3.  Section 3 should be rewritten properly, especially, lines 93 - 112. Some setences are directly from Puri et al's paper without citation. It seems that replacing $a_k = w_k^T x$ in CF and builing a neural net architecture based on CF is due to Puri et al. This paper does not mention this at all. Line 95 citation [23] for CFN seems not correct. It should be [18].  [23] is a graduate math textbook. It is highly unlikely that it contains neural networks.

Considering above, I recommends conditional acceptance on the condition that the background Section 3 is clearly rewritten. Otherwise, acceptance can be revoked.